# On Blame Attribution for Accountable Multi-Agent Sequential Decision Making

**Stelios Triantafyllou**
MPI-SWS
strianta@mpi-sws.org

**Adish Singla**
MPI-SWS
adishs@mpi-sws.org

**Goran Radanovic**
MPI-SWS
gradanovic@mpi-sws.org

## Abstract

Blame attribution is one of the key aspects of accountable decision making, as it provides means to quantify the responsibility of an agent for a decision making outcome. In this paper, we study blame attribution in the context of cooperative multi-agent sequential decision making. As a particular setting of interest, we focus on cooperative decision making formalized by Multi-Agent Markov Decision Processes (MMDPs), and we analyze different blame attribution methods derived from or inspired by existing concepts in cooperative game theory. We formalize desirable properties of blame attribution in the setting of interest, and we analyze the relationship between these properties and the studied blame attribution methods. Interestingly, we show that some of the well known blame attribution methods, such as Shapley value, are not performance-incentivizing, while others, such as Banzhaf index, may over-blame agents. To mitigate these value misalignment and fairness issues, we introduce a novel blame attribution method, unique in the set of properties it satisfies, which trade-offs explanatory power (by under-blaming agents) for the aforementioned properties. We further show how to account for uncertainty about agents' decision making policies, and we experimentally: a) validate the qualitative properties of the studied blame attribution methods, and b) analyze their robustness to uncertainty.

*... a body of people[1], holding themselves accountable to nobody, ought not to be trusted by anybody.*

*—Thomas Paine, A philosopher and a political activist.*

## 1 Introduction

With the widespread usage of artificial intelligence (AI) in everyday life [1, 2, 3], accountability has become one of the central problems in the study of AI. Much recent research studied what constitutes accountability in the context of AI and how to design accountable AI systems [4, 5, 6], and recent policies and legislations [7] are increasingly highlighting the importance of accountability, aiming to provide guidelines for developing and deploying accountable AI systems.

Accountability is a relatively broad term, and it typically involves an actor (or multiple actors) justifying their decisions and facing consequences for actions taken [6, 8]. Hence, two critical aspects of accountability are explainability and blame attribution. Recent work proposed various methods for explaining, interpreting, understanding, and certifying algorithmic decision-making and its outcomes [9, 10, 11, 12, 13, 14]. In this paper we study the other critical aspect of accountability – *blame attribution*.

---

[1]Originally, and by modern standards outdated, Thomas Paine used phrasing with the word *men*.

In multi-agent decision making, one of the central roles of blame attribution is assigning blame for undesirable outcomes or, broadly speaking, for the system's inefficiency. Prior work on responsibility and blame in AI [15, 16, 17, 18] has recognized some of the core challenges in attributing blame, including the fact that disentangling agents' contributions to the final outcome is not a trivial task. Such challenges are particularly prominent in sequential settings where past decisions influence the future ones [16].

In this paper, we consider the task of allocating a score to an agent, which represents the degree of its blame, and reflects its contributions to the total inefficiency of the multi-agent system. We focus on cooperative sequential decision making, formalized by multi-agent Markov decision processes (MMDPs) [19], where the outcome of interest is the expected discounted return of the agents' joint policy. Concretely, given an MMDP and the agents' joint policy (true or estimated), we ask: *How to score each agent so that the agents' scores satisfy desirable properties?*

To answer this question, we turn to cooperative game theory and consider blame attribution methods that are derived from or inspired by existing concepts in the cost sharing, data valuation, and coalition formation literature [20, 21, 22, 23, 24, 25, 26, 27], such as core [28], Shapley value [29, 30], or Banzhaf index [31, 32]. Taking this perspective on blame attribution, we study blame attribution for accountable multi-agent sequential decision making. More concretely:

- We formalize desirable properties that blame attribution methods should satisfy in cooperative multi-agent sequential decision making. We identify properties that are typically not considered in the cost-sharing literature, yet are important for decision making. In particular, we introduce two novel properties: a) *performance monotonicity*, which states that, having fixed all the other agents to their policies, the blame assigned to an agent should not increase if the agent adopts a policy that results in a higher expected discounted return (implying that the method is performance-incentivizing); b) *Blackstone consistency*,[2] which states that an agent should not receive a higher blame just because the agents' policies are not fully known to the blame attribution procedure.
- We characterize the properties of the studied blame attribution methods. We show that some blame assignment methods, such as, Shapley value, are not performance-monotonic (and, hence, performance-incentivizing), while others, such as Banzhaf index, may over-blame agents. Motivated by these results, we introduce a novel blame attribution method that trade-offs explanatory power (by under-blaming agents) for the aforementioned properties.
- We provide algorithms for making the studied blame attribution methods Blackstone-consistent when the agents' policies are estimated. We also characterize the effect of uncertainty on blame attribution methods.
- Using a simulation-based testbed, we experimentally analyze the studied blame attribution methods, their qualitative properties, as well as their robustness to uncertainty. The experiments showcase the importance of the robustness considerations we study and indicate that typically more efficient blame attribution methods (i.e., those that assign more blame in total) are less robust to uncertainty.

## 1.1 Other Related Work

Apart from the works mentioned in the previous paragraphs, our work relates to different areas of moral philosophy, law, and AI, and here we highlight some of the most relevant references. Research in moral philosophy and law has extensively studied the problem of blame attribution, both in terms of human actors [34, 35, 36], as well as AI actors [37, 38, 39, 40]. We take some of the well known principles in moral philosophy and law in determining properties relevant for blame attribution, e.g., Blackstone consistency is inspired by Blackstone's ratio [33]. In AI, blame attribution has been studied through a more formal lens, utilizing causality [15, 16, 17] and/or game theory [18, 41], and primarily focusing on nuances related to defining notions and degrees of responsibility, blame, and blameworthiness. In contrast, we focus on cooperative sequential decision making, and analyze how different blame attribution methods from cooperative game theory fare under different blame attribution properties. Finally, our work is generally related to the *credit assignment* problem [42, 43], and more specifically to the credit assignment problem in multi-agent reinforcement learning [44, 45, 46]. However, our focus is not on supporting the learning processes of agents by reducing computational and statistical challenges of learning, but on evaluating the agents' contributions to the system's inefficiency, ideally in a fair and interpretable manner.

---

[2]This property is inspired by Blackstone's ratio: "It is better that ten guilty persons escape than that one innocent suffer" [33].

## 2 Formal Setting

In this section, we describe our formal setting, based on multi-agent Markov decision processes (MMDPs), and we formally model the blame attribution problem in sequential decision making. This section also introduces a set of desirable formal properties of blame attribution methods.

### 2.1 Preliminaries

We consider a cooperative multi-agent setting, formalized as a class of MMDPs $\mathcal{M}$ with $n$ agents $\{1, ..., n\}$. Each MMDP in this class is a tuple $M = (\mathcal{S}, \{1, ..., n\}, \mathcal{A}, R, P, \gamma, \sigma)$ [19], where: $\mathcal{S}$ is the state space; $\mathcal{A} = \times_{i=1}^{n} \mathcal{A}_i$ is the action space, with $\mathcal{A}_i$ being the action space of agent $i$; $R$ is the reward function $R : \mathcal{S} \times \mathcal{A} \to \mathbb{R}$ specifying the reward obtained when agents $\{1, ..., n\}$ take a joint action; $P$ specifies transitions with $P(s, a, s')$ denoting the probability of transitioning to $s'$ from $s$ when agents $\{1, ..., n\}$ take joint action $a = (a_1, ..., a_n)$; $\gamma$ is the discount factor; and $\sigma$ is the initial state distribution. $\mathcal{S}$ and $\mathcal{A}$ are finite and discrete. A (stationary) joint policy $\pi$ is a mapping $\pi : \mathcal{S} \to \mathcal{D}(\mathcal{A})$, where $\mathcal{D}(\mathcal{A})$ is a probability simplex over $\mathcal{A}$, with $\pi(a|s)$ denoting the probability of taking joint action $a$ in $s$. We assume that a joint policy $\pi$ is factorizable into agents' policies, $\pi_i$, i.e., $\pi(a|s) = \pi_1(a_1|s) \cdots \pi_n(a_n|s)$. Therefore, we can define an agent $i$'s policy $\pi_i$ as a mapping from states to a distribution of agent $i$'s actions, i.e., $\pi_i : \mathcal{S} \to \mathcal{D}(\mathcal{A}_i)$. We denote the set of all policies by $\Pi = \times_{i=1}^{n} \Pi_i$. We also define a standard performance measure. The expected discounted return of a joint policy $\pi$ is defined as $J(\pi) = \mathbb{E}\left[\sum_{t=1}^{\infty} \gamma^{t-1} R(s_t, a_t)|s_1 \sim \sigma, \pi\right]$, where the initial state $s_1$ is sampled from $\sigma$, and the state-joint action pair of time-step $t$, $(s_t, a_t)$, is obtained by executing joint policy $\pi$. We abuse our notation by denoting $J(\pi_i', \pi_{-i}) = J(\pi_1, ..., \pi_i', ..., \pi_n)$. Similarly, $J(\pi_S', \pi_{-S}) = J(\pi'')$ for some $S \subseteq \{1, ..., n\}$, where $\pi_i'' = \pi_i$ if $i \notin S$ and $\pi_i'' = \pi_i'$ if $i \in S$.

### 2.2 Blame Attribution

Our goal is to assign blame to agents for failing to jointly achieve optimal performance. Given the agents' behavior policy, denoted by $\pi^b$, the inefficiency of the considered multi-agent system can be defined as $\Delta = J(\pi^*) - J(\pi^b)$, where $\pi^* \in \arg\max_\pi J(\pi)$ is an optimal joint policy. Similarly, we define the marginal inefficiency of a subset of agents $S \subseteq \{1, ..., n\}$ as $\Delta_S = J(\pi_S^{*|\pi^b}, \pi^b_{-S}) - J(\pi^b)$, as well as the marginal inefficiency of an agent $i$ as $\Delta_i = J(\pi_i^{*|\pi^b}, \pi^b_{-i}) - J(\pi^b)$, where $\pi_S^{*|\pi^b}$ (resp. $\pi_i^{*|\pi^b}$) denotes an optimal policy of $S$ (resp. $i$) assuming all other policies are fixed, i.e., $\pi_S^{*|\pi^b} \in \arg\max_{\pi_S} J(\pi_S, \pi^b_{-S})$ (resp. $\pi_i^{*|\pi^b} \in \arg\max_{\pi_i} J(\pi_i, \pi^b_{-i})$). A blame attribution method is a mapping $\Psi : \mathcal{M} \times \Pi \to \mathbb{R}_{\geq 0}^n$, where $\Psi(M, \pi^b)$ distributes blame for inefficiency $\Delta$ by assigning score $\beta_i$ to agent $i$. The output of $\Psi$, i.e., the blame assignment, is denoted by $\beta$.

**Uncertainty considerations.** Since the agents' behavior policy $\pi^b$ might not be known to the blame attribution procedure, we also define blame attribution under uncertainty as a mapping $\widehat{\Psi} : \mathcal{M} \times \mathcal{P}(\Pi) \to \mathbb{R}_{\geq 0}^n$ that outputs a blame assignment estimate $\widehat{\beta}$. Here, $\mathcal{P}(\Pi)$ represents a set whose elements express the knowledge about $\pi^b$. Inspired by the literature on robust MDPs [47, 48, 49], we encode such knowledge with uncertainty sets $\mathcal{P}(\pi^b)$, one associated to each state $s$, $\mathcal{P}(\pi^b, s)$, defined by the set of probability measures on $\mathcal{A}$. We assume that $\mathcal{P}(\pi^b)$ is consistent with $\pi^b$, i.e., $\pi^b(\cdot|s)$ is in $\mathcal{P}(\pi^b, s)$,[3,4] and that every $\pi(\cdot|s)$ in $\mathcal{P}(\pi^b, s)$ factorizes to $\pi(a|s) = \pi_1(a_1|s) \cdots \pi_n(a_n|s)$. Therefore, $\mathcal{P}(\pi^b, s)$ identifies the set of plausible stochastic actions that agent $i$ takes in state $s$.

### 2.3 Desirable Properties

Our goal is to specify functions $\Psi$ and $\widehat{\Psi}$, such that the blame assignments $\beta$ and $\widehat{\beta}$ satisfy desirable properties. In the following text we denote these properties by $\mathcal{R}$. Below, we define properties that are taken from or inspired by the game theory literature [26, 50, 21], but translated to our setting[5]:

- *Validity* ($\mathcal{R}_V$): We say that $\Psi$ is valid if it never distributes more blame than the observed inefficiency $\Delta$. More formally, $\Psi$ satisfies $\mathcal{R}_V$ (resp. $\epsilon$-$\mathcal{R}_V$) if for every $M$ and $\pi^b$, $\sum_{i=1}^{n} \beta_i \leq \Delta$ (resp. $\sum_{i=1}^{n} \beta_i \leq \Delta + \epsilon$), where $\beta = \Psi(M, \pi^b)$.

---

[3]Such $\mathcal{P}(\pi^b)$ can be derived from data containing agents' trajectories and be based on confidence intervals.
[4]$\pi^b(\cdot|s)$ could be in $\mathcal{P}(\pi^b, s)$ w.h.p., provided Blackstone consistency in Section 2.3 is similarly adjusted.
[5]Note that the terminology is slightly different.

- *Efficiency* ($\mathcal{R}_E$): A more strict condition is that the total distributed blame is equal to $\Delta$. That is, $\Psi$ satisfies $\mathcal{R}_E$ (resp. $\epsilon$-$\mathcal{R}_E$) if for every $M$ and $\pi^b$, $\sum_{i=1}^n \beta_i = \Delta$ (resp. $|\sum_{i=1}^n \beta_i - \Delta| \leq \epsilon$), where $\beta = \Psi(M, \pi^b)$.
- *Rationality* ($\mathcal{R}_R$): Similar to validity is rationality, which requires that blame distributed to any subset of agents $S$ is not greater than $\Delta_S$. That is, $\Psi$ satisfies $\mathcal{R}_R$ (resp. $\epsilon$-$\mathcal{R}_R$) if for every $M$, $\pi^b$, and $S \subseteq \{1, ..., n\}$, $\sum_{i \in S} \beta_i \leq \Delta_S$ (resp. $\sum_{i \in S} \beta_i \leq \Delta_S + \epsilon$), where $\beta = \Psi(M, \pi^b)$.
- *Symmetry* ($\mathcal{R}_S$): We say that $\Psi$ is symmetric if it treats equal agents equally, i.e., agents that equally contribute to the inefficiency should receive the same blame. More formally, $\Psi$ satisfies $\mathcal{R}_S$ (resp. $\epsilon$-$\mathcal{R}_S$) if for every $M$ and $\pi^b$, $\beta_i = \beta_j$ (resp. $|\beta_i - \beta_j| \leq \epsilon$) whenever $\Delta_{S \cup \{i\}} = \Delta_{S \cup \{j\}}$ for all $S \subseteq \{1, ..., n\} \backslash \{i, j\}$, where $\beta = \Psi(M, \pi^b)$.
- *Invariance* ($\mathcal{R}_I$): We say that $\Psi$ is invariant if it assigns zero blame to agents who do not marginally contribute to inefficiency. More formally, $\Psi$ satisfies $\mathcal{R}_I$ (resp. $\epsilon$-$\mathcal{R}_I$) if for every $M$ and $\pi^b$, $\beta_i = 0$ (resp. $\beta_i \leq \epsilon$) whenever $\Delta_{S \cup \{i\}} = \Delta_S$ for all $S$, where $\beta = \Psi(M, \pi^b)$.

Note that $\epsilon > 0$ in the definitions of $\epsilon$-$\mathcal{R}$, and that we use these properties in our characterization result for blame attribution under uncertainty. Additionally, we consider two properties that relate the blame attribution output to the MMDP structure and the agents' behavior policies.

- *Contribution monotonicity* ($\mathcal{R}_{CM}$)[51]: We say that $\Psi$ is contribution-monotonic if the blame it assigns to an agent depends only on its marginal contributions and monotonically so. More formally, $\Psi$ satisfies $\mathcal{R}_{CM}$ (resp. $\epsilon$-$\mathcal{R}_{CM}$) if for every two $(M^1, \pi^{b^1})$ and $(M^2, \pi^{b^2})$, $\beta_i^1 \geq \beta_i^2$ (resp. $\beta_i^1 \geq \beta_i^2 - \epsilon$) whenever $\Delta_{S \cup \{i\}}^1 - \Delta_S^1 \geq \Delta_{S \cup \{i\}}^2 - \Delta_S^2$ for all $S$, where $\beta^1 = \Psi(M^1, \pi^{b^1})$ and $\beta^2 = \Psi(M^2, \pi^{b^2})$.
- *Performance monotonicity* ($\mathcal{R}_{PerM}$): We say that $\Psi$ is performance-monotonic if it does not assign greater blame to agent $i$ for adopting a policy that results in an equal or higher performance, assuming the other agents' policies fixed. More formally, consider any MMDP $M$, and any $\pi^b_{-i}$, $\pi_i$ and $\pi_i'$ such that $J(\pi_i, \pi^b_{-i}) \leq J(\pi_i', \pi^b_{-i})$. We say that $\Psi$ satisfies $\mathcal{R}_{PerM}$ (resp. $\epsilon$-$\mathcal{R}_{PerM}$) if $\beta_i \geq \beta_i'$ (resp. $\beta_i \geq \beta_i' - \epsilon$) where $\beta = \Psi(M, (\pi_i, \pi^b_{-i}))$ and $\beta' = \Psi(M, (\pi_i', \pi^b_{-i}))$.

The above definitions directly extend to $\widehat{\Psi}$ except that we require them to hold for all $\mathcal{P}(\pi_b)$. Additionally, we identify the following property for $\widehat{\Psi}$:

- *Blackstone consistency* ($\mathcal{R}_{BC}$): We say that $\widehat{\Psi}$ is Blackstone-consistent with $\Psi$ if it never attributes more blame to an agent than $\Psi$. More formally, $\widehat{\Psi}$ satisfies $\mathcal{R}_{BC}(\Psi)$ if for any $M$, $\pi^b$ and $\mathcal{P}(\pi^b)$, $\widehat{\beta}_i \leq \beta_i$, where $\beta = \Psi(M, \pi^b)$ and $\widehat{\beta} = \widehat{\Psi}(M, \mathcal{P}(\pi^b))$.

## 3 Game-Theoretic Approaches to Blame Attribution

In this section, we study blame attribution methods based on well known game theoretic notions, such as the core [28], Shapley value [29, 30], or Banzhaf index [31, 32]. We also introduce a novel blame attribution method, unique in the set of properties it satisfies. The proofs of our results can be found in the supplementary material.

### 3.1 Max-Efficient Rationality

We start with a relatively simple blame assignment method that puts rationality as a strict condition, and maximizes the efficiency of blame assignment under this constraint. We call this blame assignment method *max-efficient rationality*. More formally, max-efficient rationality can be defined via the following linear program:

$$\Psi_{MER}(M, \pi^b) := \max_\beta \sum_{i=1}^n \beta_i \qquad \text{s.t.} \quad \sum_{i \in S} \beta_i \leq \Delta_S \quad \forall S \subseteq \{1, ..., n\}, \qquad \text{(P1)}$$

where $\Delta_S$ are precomputed. Max-efficient rationality is inspired by the notion of core, but unlike the core, max-efficient rationality does not require $\mathcal{R}_E$ (efficiency) to hold. It is easy to show that the following properties are satisfied by any optimizer of (P1), $\Psi_{MER}$.

**Proposition 1.** *Every solution to the optimization problem* (P1)*, i.e., $\Psi_{MER}$, satisfies $\mathcal{R}_V$ (validity), $\mathcal{R}_R$ (rationality) and $\mathcal{R}_I$ (invariance).*

Since there might exist multiple optimal solutions to (P1), a tie breaking rule might be needed to decide on the method's output, $\Psi_{MER}$. We account for this fact in the experiments from Section 5. Note that the constraints in (P1) are quite restrictive, leading to blame assignments that typically distribute very little blame in total. The amount of total blame assigned is important for explanatory power. Namely, a trivial blame attribution method that assigns the score of 0 to every agent satisfies all of the properties from the previous section except $\mathcal{R}_E$ (efficiency), but provides no information regarding the agents' contributions to the outcome.

### 3.2 Marginal Contribution

Another intuitive blame assignment method is what we refer to as *marginal contribution*. This method simply quantifies an agent's potential to increase the performance of the system, assuming that the other agents keep their policies fixed. That is, the blame assigned to agent $i$ is equal to $\beta_i = \Delta_i$. The following properties hold:

**Proposition 2.** $\Psi_{MC}(M, \pi^b) = (\Delta_1, ..., \Delta_n)$ *satisfies* $\mathcal{R}_S$ *(symmetry),* $\mathcal{R}_I$ *(invariance),* $\mathcal{R}_{CM}$ *(contribution monotonicity) and* $\mathcal{R}_{PerM}$ *(performance monotonicity).*

Unlike max-efficient rationality, marginal contribution does not satisfy validity, i.e., it can over-blame a group of agents by assigning them total score that exceeds the improvement they can achieve, i.e., $\Delta$. Given that an agent's marginal inefficiency is not always a good indicator of the agent's influence on the system's performance, this method can be highly inefficient (distributing very little blame) when coordination among agents is required, as we show in Section 5.

### 3.3 Shapley Value and Banzhaf Index

In the context of the sequential decision making setting studied in this paper, Shapley value can be defined as $\beta = \Psi_{SV}(M, \pi^b)$ such that

$$\beta_i = \sum_{S \subseteq \{1,...,n\} \setminus \{i\}} w_S \cdot \left[ J(\pi_{S \cup \{i\}}^{*|\pi^b}, \pi^b_{-S \cup \{i\}}) - J(\pi_S^{*|\pi^b}, \pi^b_{-S}) \right], \tag{1}$$

where coefficients $w_S$ are set to $w_S = \frac{|S|!(n-|S|-1)!}{n!}$. We restate (and in the supplementary material, prove the claim for our setting) a well known uniqueness result for Shapley value:

**Theorem 1.** *[51]* $\Psi_{SV}(M, \pi^b) = (\beta_1, ..., \beta_n)$, *where* $\beta_i$ *is defined by Eq. (1) and* $w_S = \frac{|S|!(n-|S|-1)!}{n!}$, *is a unique blame attribution method satisfying* $\mathcal{R}_E$ *(efficiency),* $\mathcal{R}_S$ *(symmetry) and* $\mathcal{R}_{CM}$ *(contribution monotonicity). Additionally,* $\Psi_{SV}$ *satisfies* $\mathcal{R}_V$ *(validity) and* $\mathcal{R}_I$ *(invariance).*

As we show in Section 5, Shapley value does not satisfy properties $\mathcal{R}_R$ (rationality) nor $\mathcal{R}_{PerM}$ (performance monotonicity).

Banzhaf index, denoted by $\Psi_{BI}$, is similar to Shapley value, and in fact, it has the same functional form but different coefficients ($w_S = \frac{1}{2^{n-1}}$), leading to a slightly different uniqueness result. The supplementary material discusses Banzhaf index and its properties in greater detail. Here, we note that Banzhaf index is equivalent to Shapley value for two agents. However, in general, Banzhaf index does not satisfy $\mathcal{R}_E$ (efficiency), but a version of it, called 2-efficiency [52]. As it is the case with Shapley value, Banzhaf index does not satisfy $\mathcal{R}_{PerM}$ (performance monotonicity) nor $\mathcal{R}_R$ (rationality). Interestingly, $\mathcal{R}_V$ (validity) might also not hold (see Section 5).

### 3.4 Average Participation

Motivated by the fact that $\mathcal{R}_{PerM}$ (performance monotonicity) is important for incentivizing good performance and $\mathcal{R}_E$ (efficiency) is important for explanatory power, we introduce a novel blame assignment method, which can be seen as a combination of marginal contribution and Shapley value. We first show the following result, which shows that there is an inherent trade-off between $\mathcal{R}_{PerM}$ and $\mathcal{R}_E$, assuming $\mathcal{R}_S$ (symmetry) and $\mathcal{R}_I$ (invariance) hold.

**Proposition 3.** *No blame attribution method* $\Psi$ *satisfies* $\mathcal{R}_E$ *(efficiency),* $\mathcal{R}_S$ *(symmetry),* $\mathcal{R}_I$ *(invariance) and* $\mathcal{R}_{PerM}$ *(performance monotonicity).*

Given this result, we instead consider two new properties $\mathcal{R}_{AE}$ (average efficiency) and $\mathcal{R}_{cPerM}$ (c-performance monotonicity), which are weaker variants of $\mathcal{R}_E$ (efficiency) and $\mathcal{R}_{PerM}$ (performance

monotonicity) respectively. Importantly, $\mathcal{R}_{AE}$ is not satisfied by $\Psi_{MC}$ and $\mathcal{R}_{cPerM}$ by $\Psi_{SV}$. In addition, we also consider two variants of $\mathcal{R}_{CM}$ (contribution monotonicity): $\mathcal{R}_{cParM}$ (c-participation monotonicity) and $\mathcal{R}_{RcParM}$ (relative c-participation monotonicity). To define the new properties, we introduce a contribution function $c : \mathcal{M} \times \Pi \times \{1, ..., n\} \rightarrow \{0, 1\}$ that indicates whether an agent is *pivotal*, i.e., marginally contributes to the inefficiency of some subset of $\{1, ..., n\}$:

$$c(M, \pi^b, i) = \begin{cases} 0 & \text{if } J(\pi^{*|\pi^b}_{S \cup \{i\}}, \pi^b_{-S \cup \{i\}}) = J(\pi^{*|\pi^b}_{S}, \pi^b_{-S}) \quad \forall S \subseteq \{1, ..., n\} \\ 1 & \text{otherwise} \end{cases}.$$

Alternatively, an agent $i$ is pivotal if and only if its Shapley value is strictly greater than 0, i.e., $c(M, \pi^b, i) = \mathbb{1}[\beta_i > 0]$, where $\beta = \Psi_{SV}(M, \pi^b)$ and $\mathbb{1}[.]$ is an indicator function. The new properties are then defined as follows:

- *Average efficiency* ($\mathcal{R}_{AE}$): $\Psi$ satisfies $\mathcal{R}_{AE}$ (resp. $\epsilon\text{-}\mathcal{R}_{AE}$) if for every $M$ and $\pi^b$, $\sum_{i=1}^{n} \beta_i = \sum_{S \subseteq \{1,...,n\}} \frac{1}{2^n - 1} \cdot \Delta_S$ (resp. $|\sum_{i=1}^{n} \beta_i - \sum_{S \subseteq \{1,...,n\}} \frac{1}{2^n - 1} \cdot \Delta_S| \leq \epsilon$), where $\beta = \Psi(M, \pi^b)$.
- *c-Performance monotonicity* ($\mathcal{R}_{cPerM}$): Consider any MMDP $M$, and any $\pi^b_{-i}$, $\pi_i$ and $\pi'_i$ s.t. $J(\pi_i, \pi^b_{-i}) \leq J(\pi'_i, \pi^b_{-i})$ and $c(M, (\pi_i, \pi^b_{-i}), j) = c(M, (\pi'_i, \pi^b_{-i}), j)$ for every $j$. We say that $\Psi$ satisfies $\mathcal{R}_{cPerM}$ (resp. $\epsilon\text{-}\mathcal{R}_{cPerM}$) if $\beta_i \geq \beta'_i$ (resp. $\beta_i \geq \beta'_i - \epsilon$) where $\beta = \Psi(M, (\pi_i, \pi^b_{-i}))$ and $\beta' = \Psi(M, (\pi'_i, \pi^b_{-i}))$.
- *c-Participation monotonicity* ($\mathcal{R}_{cParM}$): $\Psi$ satisfies $\mathcal{R}_{cParM}$ (resp. $\epsilon\text{-}\mathcal{R}_{cParM}$) if for every $(M^1, \pi^{b^1})$ and $(M^2, \pi^{b^2})$ s.t. $c(M^1, \pi^{b^1}, i) = c(M^2, \pi^{b^2}, i)$ for every $i$, $\beta_j^{\,1} \geq \beta_j^{\,2}$ (resp. $\beta_j^{\,1} \geq \beta_j^{\,2} - \epsilon$) whenever $\Delta^1_{S \cup \{j\}} \geq \Delta^2_{S \cup \{j\}}$ for all $S$, where $\beta^1 = \Psi(M^1, \pi^{b^1})$ and $\beta^2 = \Psi(M^2, \pi^{b^2})$.
- *Relative c-participation monotonicity* ($\mathcal{R}_{RcParM}$): $\Psi$ satisfies $\mathcal{R}_{RcParM}$ (resp. $\epsilon\text{-}\mathcal{R}_{RcParM}$) if for every $(M^1, \pi^{b^1})$ and $(M^2, \pi^{b^2})$ s.t. $c(M^1, \pi^{b^1}, i) = c(M^2, \pi^{b^2}, i)$ for every $i$, $\beta_j^{\,1} - \beta_j^{\,2} \geq \beta_k^{\,1} - \beta_k^{\,2}$ (resp. $\beta_j^{\,1} - \beta_j^{\,2} \geq \beta_k^{\,1} - \beta_k^{\,2} - \epsilon$) whenever $c(M^1, \pi^{b^1}, j) = c(M^1, \pi^{b^1}, k)$ and $\Delta^1_{S \cup \{j\}} - \Delta^2_{S \cup \{j\}} \geq \Delta^1_{S \cup \{k\}} - \Delta^2_{S \cup \{k\}}$ for all $S \in \{1, ..., n\} \backslash \{j, k\}$, where $\beta^1 = \Psi(M^1, \pi^{b^1})$ and $\beta^2 = \Psi(M^2, \pi^{b^2})$.

Before describing the main results of this subsection, we briefly outline the intuition behind the above definitions. $\mathcal{R}_{AE}$ (average efficiency) is similar to $\mathcal{R}_E$ (efficiency), however it requires less total blame to be distributed. Whereas $\mathcal{R}_E$ requires that the total blame is equal to the total inefficiency $\Delta$, $\mathcal{R}_{AE}$ requires that the total blame is equal to the average marginal inefficiency of subsets of agents, i.e., the average value of $\Delta_S$.[6] Compared to $\mathcal{R}_{PerM}$ (performance monotonicity), $\mathcal{R}_{cPerM}$ (c-performance monotonicity) additionally accounts for the pivotality of agents through contribution function $c$, treating each set of pivotal agents as a separate case. $\mathcal{R}_{cParM}$ (c-participation monotonicity) accounts for agents' pivotality in a similar manner. Moreover, $\mathcal{R}_{cParM}$ resembles contribution monotonicity $\mathcal{R}_{CM}$, but instead of requiring blame monotonicity to hold w.r.t. the agent's influence on the marginal inefficiency of subsets $S$ (i.e., $\Delta_{S \cup \{i\}} - \Delta_S$), it considers blame monotonicity w.r.t. the marginal inefficiency of subsets that contain the agent (i.e., $\Delta_{S \cup \{j\}}$). Relative c-participation monotonicity $\mathcal{R}_{RcParM}$ is similar to $\mathcal{R}_{cParM}$, but its blame monotonicity requirement is based on a pairwise comparison of agents with the same pivotality degree. In particular, $\mathcal{R}_{RcParM}$ requires that the blame increase is higher for an agent who is in subsets with a greater marginal inefficiency increase (i.e., $\beta_j^{\,1} - \beta_j^{\,2} \geq \beta_k^{\,1} - \beta_k^{\,2}$ whenever $\Delta^1_{S \cup \{j\}} - \Delta^2_{S \cup \{j\}} \geq \Delta^1_{S \cup \{k\}} - \Delta^2_{S \cup \{k\}}$).

**Average participation**: Now, we describe the new blame assignment method, which we call *average participation*. This blame assignment method can be defined as $\beta = \Psi_{AP}(M, \pi^b)$ such that

$$\beta_i = \sum_{S \subseteq \{1,...,n\} \backslash \{i\}} w \cdot \frac{c(M, \pi^b, i)}{\sum_{j \in S} c(M, \pi^b, j) + 1} \cdot \Delta_{S \cup \{i\}}, \tag{2}$$

where coefficient $w$ is set to $w = \frac{1}{2^n - 1}$. Intuitively, $\Psi_{AP}$ equally distributes blame for the marginal inefficiency of a subset of agents among the pivotal agents in that subset. Hence, an agent $i$ that is pivotal receives blame for each subset $S \cup \{i\}$ equal to $\Delta_{S \cup \{i\}}$ divided by the number of pivotal agents in $S \cup \{i\}$ and scaled by coefficient $w$. Agents that are not pivotal, obtain 0 blame. Average participation uniquely satisfies the following properties.

---

[6]This average does not include $\Delta_\emptyset$, which is equal to 0. Note also that $\Delta \geq \Delta_S$ for every $S \subseteq \{1, ..., n\}$, so this average is upper bounded by $\Delta$.

**Theorem 2.** $\Psi_{AP}(M, \pi^b) = (\beta_1, ..., \beta_n)$, where $\beta_i$ is defined by Eq. (2) and $w = \frac{1}{2^n-1}$, is a unique blame attribution method that satisfies $\mathcal{R}_{AE}$ (average-efficiency), $\mathcal{R}_S$ (symmetry), $\mathcal{R}_I$ (invariance), $\mathcal{R}_{cParM}$ (c-participation monotonicity) and $\mathcal{R}_{RcParM}$ (relative c-participation monotonicity). Furthermore, $\Psi_{AP}$ satisfies $\mathcal{R}_{cPerM}$ (c-performance monotonicity) and $\mathcal{R}_V$ (validity).

Unlike marginal contribution, average participation is valid (never over-blames agents), however it satisfies a weaker version of performance monotonicity. Still, this version is not satisfied by Shapley value. On the other hand, Shapley value is efficient, unlike average participation, which satisfies a weaker requirement—average efficiency. We also showcase these trade-offs in Section 5.

## 4 Blame Attribution under Uncertainty

In this section, we study blame attribution methods that do not have direct access to $\pi^b$. As mentioned in Section 2, we focus on the case where the knowledge about $\pi^b$ is defined by the uncertainty set $\mathcal{P}(\pi^b)$, and it is defined state-wise so that each state is associated with a set of probability measures on $\mathcal{A}$ identifying plausible candidates for $\pi^b(\cdot|s)$.[7] We denote $\pi \in \mathcal{P}(\pi^b)$ if $\pi$ is plausible by $\mathcal{P}(\pi^b)$.

### 4.1 Shapley Value under Uncertainty

In explaining approaches to handling uncertainty, we focus on Shapley value. Arguably, the simplest way to operate under uncertainty is to derive a point estimate of $\pi^b$, denoted by $\widehat{\pi}^b$,[8] and apply $\Psi_{SV}$ on this estimate to obtain blame assignment $\widehat{\beta} = \Psi_{SV}(M, \widehat{\pi}^b)$. Albeit being simple, this approach does not satisfy desirable properties, most notably, $\mathcal{R}_V$ (validity) and $\mathcal{R}_{BC}$ (Blackstone consistency).

**Validity.** Now, note that $\widehat{\beta} = \Psi_{SV}(M, \widehat{\pi}^b)$ satisfies $\sum_{i=1}^n \widehat{\beta}_i = J(\pi^*) - J(\widehat{\pi}^b)$. Therefore, instead of relying on a point estimate $\widehat{\pi}^b$, we could utilize a policy $\widehat{\pi}^b$ for which $J(\pi^*) - J(\widehat{\pi}^b) \leq \Delta$. Namely, in that case $\widehat{\beta} = \Psi_{SV}(M, \widehat{\pi}^b)$ results in a blame assignment that satisfies $\mathcal{R}_V$ (validity). Since this inequality holds for a solution to the optimization problem $\max_{\pi \in \mathcal{P}(\pi^b)} J(\pi)$, we obtain:

**Proposition 4.** Let $\widehat{\pi}^b$ be a solution to the optimization problem $\max_{\pi \in \mathcal{P}(\pi^b)} J(\pi)$. Then $\widehat{\Psi}_{SV,V}(M, \mathcal{P}(\pi^b)) = \Psi_{SV}(M, \widehat{\pi}^b)$ satisfies $\mathcal{R}_V$ (validity).

**Blackstone consistency.** As we show in Section 5, although $\widehat{\Psi}_{SV,V}$ is valid, it might not be Blackstone consistent w.r.t. $\Psi_{SV}$. In particular, although the total blame is never overestimated, an agent $i$ might receive higher blame than it would receive under $\Psi_{SV}$. To ensure Blackstone consistency, we can assign blame to agent $i$ equal to $\min_{\pi \in \mathcal{P}(\pi^b)} \beta_i^{\pi}$ s.t. $\beta^{\pi} = \Psi_{SV}(M, \pi)$. Together with Eq. (1), this implies that agent $i$'s blame is obtained by solving

$$\min_{\pi \in \mathcal{P}(\pi^b)} \sum_{S \subseteq \{1,...,n\} \setminus \{i\}} w_S \cdot \left[ J(\pi_{S \cup \{i\}}^{*|\pi}, \pi_{-S \cup \{i\}}) - J(\pi_S^{*|\pi}, \pi_{-S}) \right], \tag{P2}$$

where $w_S = \frac{|S|!(n-|S|-1)!}{n!}$ and $\pi_S^{*|\pi} \in \arg\max_{\pi'_S} J(\pi'_S, \pi_{-S})$. We have the following result:

**Proposition 5.** Let $\beta_i^i$ be the minimum value of the objective in (P2). Then $\widehat{\Psi}_{SV,BC}(M, \mathcal{P}(\pi^b)) = (\beta_1^1, ..., \beta_n^n)$ satisfies $\mathcal{R}_V$ (validity) and $\mathcal{R}_{BC}(\Psi_{SV})$ (Blackstone consistency w.r.t. $\Psi_{SV}(M, \pi^b)$).

Note that $\widehat{\Psi}_{SV,BC}$ distributes less total blame than $\widehat{\Psi}_{SV,V}$, since it takes the worst case perspective for each agent separately, while under $\widehat{\Psi}_{SV,V}$ the blame assigned to all agents is computed with the same joint behavior policy. Moreover, the objective function in (P2) is more complex than in classical robust MDP settings [47, 48], making classical approaches for robust MDPs hard to apply. In practice, we can relax (P2) and optimize a lower bound of the objective; this preserves $\mathcal{R}_{BC}(\Psi_{SV})$, but at the expense of distributing less blame to the agents. In our experiments, we solve $\min_{\pi \in \mathcal{P}'(\pi^b)} J(\pi_{S \cup \{i\}}^{*|\pi}, \pi_{-S \cup \{i\}})$ and $\max_{\pi \in \mathcal{P}'(\pi^b)} J(\pi_S^{*|\pi}, \pi_{-S})$ for each subset $S$ and with appropriately chosen $\mathcal{P}'(\pi^b) \supseteq \mathcal{P}(\pi^b)$ (see the supplementary material), and we apply Eq. (1) to

---

[7]Such definition implies a rectangularity of the uncertainty set [47, 49].

[8]For example, this estimate can be derived from data containing the agents' trajectories.

obtain the blame assignment. This implies that agent $i$'s blame is obtained by solving

$$\sum_{S \subseteq \{1,...,n\} \backslash \{i\}} w_S \cdot \left[ \min_{\pi \in \mathcal{P}'(\pi^b)} J(\pi^{*|\pi}_{S \cup \{i\}}, \pi_{-S \cup \{i\}}) - \max_{\pi \in \mathcal{P}'(\pi^b)} J(\pi^{*|\pi}_{S}, \pi_{-S}) \right].$$

**Other Blame Attribution Methods.** Similar approaches also work for other blame assignment methods discussed in Section 3. For example, and focusing on Blackstone consistency, $\widehat{\Psi}_{BI,BC}(M, \mathcal{P}(\pi^b))$ can be obtained in the same way as $\widehat{\Psi}_{SV,BC}(M, \mathcal{P}(\pi^b))$, but with $w_S = \frac{1}{2^{n-1}}$, while $\widehat{\Psi}_{MC,BC}(M, \mathcal{P}(\pi^b))$ can be implemented as $\widehat{\Psi}_{MC,BC}(M, \mathcal{P}(\pi^b)) = (\tilde{\Delta}_1, ..., \tilde{\Delta}_n)$ where $\tilde{\Delta}_i = \min_{\pi \in \mathcal{P}'(\pi^b)} J(\pi^{*|\pi}_i, \pi_{-i}) - \max_{\pi \in \mathcal{P}'(\pi^b)} J(\pi)$. Implementing Blackstone consistent $\widehat{\Psi}_{MER,BC}(M, \mathcal{P}(\pi^b))$ and $\widehat{\Psi}_{AP,BC}(M, \mathcal{P}(\pi^b))$ is more nuanced, and we discuss it in the supplementary material.

## 4.2 Characterization Result

Notice that the described Blackstone consistent methods $\widehat{\Psi}(M, \mathcal{P}(\pi^b))$ are not guaranteed to satisfy the properties that their counterparts $\Psi(M, \pi^b)$ satisfy. However, as long as $\widehat{\Psi}(M, \mathcal{P}(\pi^b))$ and $\Psi(M, \pi^b)$ output similar enough blame assignments, properties that hold under $\Psi(M, \pi^b)$ will approximately hold under $\widehat{\Psi}(M, \mathcal{P}(\pi^b))$. More formally, we have the following results.

**Theorem 3.** *Consider $\widehat{\Psi}$ and $\Psi$ s.t. $\left\| \widehat{\Psi}(M, \mathcal{P}(\pi^b)) - \Psi(M, \pi^b) \right\|_1 \leq \epsilon$ for any $M$, $\pi^b$, and $\mathcal{P}(\pi^b)$. Then if $\Psi$ satisfies a property $\mathcal{R} \in \{\mathcal{R}_V, \mathcal{R}_E, \mathcal{R}_R, \mathcal{R}_S, \mathcal{R}_I, \mathcal{R}_{AE}\}$, $\widehat{\Psi}$ satisfies $\epsilon$-$\mathcal{R}$. Moreover, if $\Psi$ satisfies a property $\mathcal{R} \in \{\mathcal{R}_{CM}, \mathcal{R}_{PerM}, \mathcal{R}_{cPerM}, \mathcal{R}_{cParM}, \mathcal{R}_{RcParM}\}$, $\widehat{\Psi}$ satisfies $2\epsilon$-$\mathcal{R}$.*

This theorem allows us to quantify the robustness of the blame attribution methods—the closer $\widehat{\Psi}$ is to $\Psi$, the more robust it is to uncertainty. Interestingly, a trivial blame attribution method that assigns 0 blame to all the agents is robust in this sense. However, as we already mentioned, this trivial blame assignment is not informative as it does not attribute any blame. In fact, if agents receive no penalties for bad behavior, such a blame attribution method might have adverse effects. We provide a broader discussion on the negative side-effects of under-blaming in the supplementary material. Importantly, this example suggests that efficiency (in a broad sense, i.e., how much blame is being distributed) and robustness are sometimes at odds, which we also demonstrate in the experiments.

## 5  Experiments

To demonstrate the efficacy of the studied blame attribution methods, we consider two environments, *Gridworld* and *Graph*, depicted in Fig. 1 and Fig. 2. Both environments are adapted from [53] and modified to be multi-agent. The experiments evaluate blame attribution methods along three axis:

- *Performance monotonicity:* First, we test blame attribution methods for the $\mathcal{R}_{PerM}$ (performance monotonicity) property, which we deem important for accountability. To do that, we consider the gridworld environment: this is a two-agent environment in which one of the agents, $A_2$, optimizes its policy using a model of the other agent, $A_1$. Importantly, by controlling the correctness of $A_2$'s model of $A_1$, we can validate whether a blame attribution method satisfies $\mathcal{R}_{PerM}$. Namely, if $A_2$ does not receive the minimum blame when its model of $A_1$ is the correct model, the corresponding method is not performance incentivizing, i.e., it does not satisfy $\mathcal{R}_{PerM}$.
- *Coordination:* Second, we evaluate the efficacy of blame attribution methods when a higher degree of coordination among agents is needed to yield improvements over the baseline behavior. For this, we consider the graph environment, which includes configurations where an agent cannot improve the joint performance by unilaterally changing its policy. Thus, this environment is suitable for evaluating whether blame attribution methods incorporate more nuanced counterfactual reasoning.
- *Robustness:* Finally, we evaluate the robustness of blame attribution methods under uncertainty. In this case, both environments (Gridworld and Graph) are used for testing purposes, and we control for the level of uncertainty over the agents' behavior policies.

The supplementary material provides more details on the experimental setup and implementation. Below we provide a more detailed description of the considered environments and discuss our findings.

**Environment 1**: This is a gridworld environment, in which two agents control the same actor but with different priorities. In the single-agent version of the environment, an agent, agent $A_1$, controls the movement of the actor. In our multi-agent version, there is an additional agent, agent $A_2$, who can intervene and override $A_1$'s actions. The two agents select their actions simultaneously. Cells denoted with $S$ are the initial states, blank cells indicate areas of small negative reward, $F$ cells indicate areas of slightly increased cost and $H$ cells are areas of severe penalty. The cell denoted by $G$ is the terminal state of the environment and has a positive reward. When agent $A_2$ intervenes in some state, the actor takes the action that an optimal policy would select in the single-agent mode, but also pays a cost of

Fig. 1. Gridworld

intervention $C$. The behavior policy $\pi^b_1$ of agent $A_1$ is parameterized by variable $\alpha$, which specifies the probability that $A_1$ takes an action determined by an optimal single-agent policy, instead of its personal policy. The personal policy of $A_1$ is a mixture of an optimal single-agent policy for correctly specified costs and a single-agent policy that is optimal but for misspecified costs of $F$ and $H$ cells—it assumes that they have the same cost as the blank cells. $A_2$'s behavior policy $\pi^b_2$ optimizes the expected discounted return and is trained with a model of $A_1$ specified by the true personal policy of $A_1$ and variable $\alpha'$ (not necessarily equal to $\alpha$). $A_2$ is meant to rectify potential mistakes of $A_1$ that could inflict cost greater than $C$. In $\mathcal{R}_{PerM}$ experiments we set $\alpha = 0.4$. In robustness experiments, we only consider uncertainty over the personal policy of $A_1$, and we set $\alpha = 0.2$ and $\alpha' = 0.5$.

**Performance monotonicity**: Fig. 3a validates our theoretical results regarding $\mathcal{R}_{PerM}$ (performance monotonicity). More specifically, methods $\Psi_{AP}$ and $\Psi_{MC}$ assign the minimum blame to $A_2$ when it acts optimally w.r.t. the true policy of $A_1$, i.e., when $\alpha' = \alpha$. However, this is not the case for methods $\Psi_{SV}$ and $\Psi_{BI}$, which implies that these methods are not incentivizing $A_2$ to act optimally w.r.t. its belief about $A_1$. $\Psi_{MER}$ and $\Psi_{BI}$ assign the same blame to $A_2$ as $\Psi_{MC}$ and $\Psi_{SV}$, respectively.

**Environment 2**: This is a graph environment in which 4 agents simultaneously select actions. The graph consists of one starting and one terminal node, as well as 8 intermediate nodes that can be grouped according to their index number; nodes with even index number are located on the upper level of the graph and nodes with odd index number on the lower level. At each time-step every agent

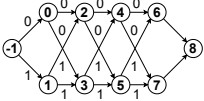

Fig. 2. Graph

chooses to take either action $0$ and move to the upper level or action $1$ and move to the lower level. We test multiple variants of this environment, each of which defines a different reward function. In all variants, the reward at each time-step is $+1$ if some formation constraint is satisfied and $-1$ if not. In the first set of experiments (Coordination), we consider 4 different formation constraints: in formation constraint $m \in \{1, ..., 4\}$, at least $m$ agents need to select action $1$ for the constraint to be satisfied. Each behavior policy $\pi^b_i$ takes action $a_i = 0$ in every node. In the second set of experiments (Robustness), we consider one formation constraint that is satisfied if the agents are arranged equally between the two levels. In states where agents are balanced between the levels, each behavior policy $\pi^b_i$ takes the action from the previous time-step with probability $p_i$; in unbalanced states, the action that leads to the level with the least number of agents is taken with probability $p_i$.

**Coordination**: Fig. 3e shows how much blame in total the blame attribution methods assign for the four different levels of required coordination ($m = 1, ..., 4$). Observe, that when the constraint can be satisfied by every agent ($m = 1$), $\Psi_{MC}$ violates $\mathcal{R}_V$ (validity). For $m = 2$, $\Psi_{MER}$ and $\Psi_{MC}$ assign zero blame to all agents, while $\Psi_{BI}$ violates $\mathcal{R}_V$ (validity). Although always valid, $\Psi_{AP}$ assigns significantly less blame as $m$ increases. $\Psi_{SV}$ is always efficient, and its total blame does not vary with $m$. $\Psi_{SV}$, $\Psi_{BI}$, $\Psi_{MC}$ and $\Psi_{AP}$ do not satisfy $\mathcal{R}_R$ (they assign more total blame than $\Psi_{MER}$).

**Robustness**: We test the robustness of the blame attribution methods by controlling the amount of uncertainty in the estimates of the agents' behavior policies. To model uncertainty, we consider maximum estimation error $\epsilon_{max}$, and to obtain uncertainty sets $\mathcal{P}(\pi^b)$, we sample (uniformly at random) $\widehat{\pi}^b_i(s)$ such that $\frac{1}{2}\left\|\widehat{\pi}^b_i(s) - \pi^b_i(s)\right\|_1 \le \epsilon_{max}$. Moreover, $\mathcal{P}(\pi^b, s)$ contains all policies $\pi$ such that $\frac{1}{2}\left\|\widehat{\pi}^b_i(s) - \pi_i(s)\right\|_1 \le \epsilon_{max}$. In our experiments, we take $\widehat{\pi}^b$ to be the point estimate of $\pi^b$.

*Comparing estimation approaches*: Fig. 3b and 3f show how the approaches for estimating $\widehat{\Psi}_{SV}$ from Section 4 fare under different levels of uncertainty. The point estimate approach typically over-blames an agent and the amount of over-blaming increases with the level of uncertainty. $\widehat{\Psi}_{SV,BC}$ never over-blames any agent, but it becomes less efficient (in distributing blame) as $\epsilon_{max}$ increases (Fig. 3f). $\widehat{\Psi}_{SV,V}$ is more efficient than $\widehat{\Psi}_{SV,BC}$, but violates $\mathcal{R}_{BC}$ (Blackstone consistency) (Fig. 3b).

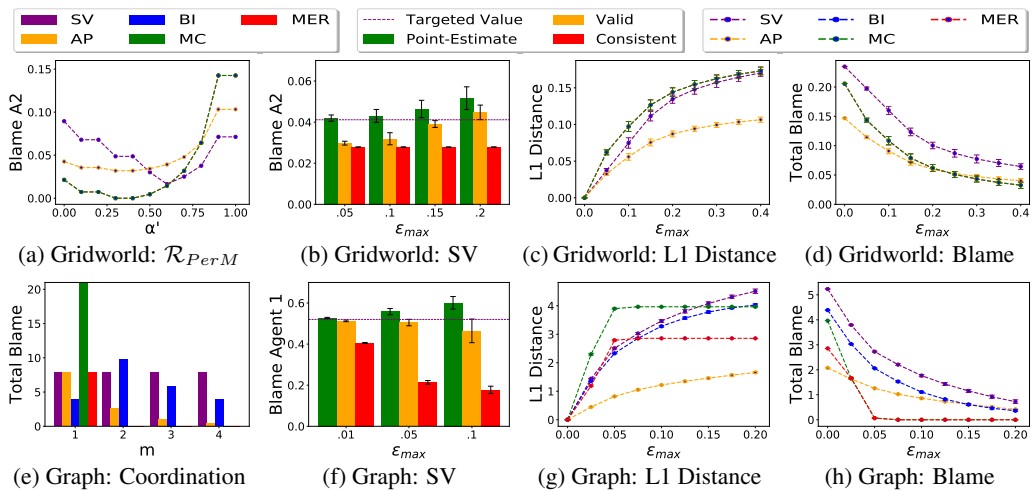

Fig. 3. Experimental results for the Gridworld and Graph environments. Plot (a) tests methods for $\mathcal{R}_{PerM}$. Plot (e) shows the effect of varying coordination level. Plots (b,c,d,f,g,h) show the effect of varying $\epsilon_{max}$ in different Shapley value approaches (b,f) and blame attribution methods (c,d,g,h).

*Comparing attribution approaches*: Fig. 3c and 3g show for each consistent blame attribution method $\widehat{\Psi}$ from Section 4 the $L_1$ distance between its output and the output of its counterpart $\Psi$ ("targeted assignment"). Fig. 3d and 3h show the total blame assigned by these methods. $\widehat{\Psi}_{AP,BC}$ consistently outperforms the other methods in terms of the $L_1$ distance from its "targeted assignment". Compared to $\widehat{\Psi}_{AP,BC}$, $\widehat{\Psi}_{SV,BC}$ is consistently better in terms of efficiency (in distributing blame). Similar, albeit less prominent effects can be seen when comparing $\widehat{\Psi}_{AP,BC}$ and $\widehat{\Psi}_{BI,BC}$. These results indicate a tendency where efficiency (in distributing blame) and robustness are at odds, as we also discuss in Section 4.2. $\widehat{\Psi}_{MER,BC}$ and $\widehat{\Psi}_{MC,BC}$ assign zero total blame even for smaller $\epsilon_{max}$, indicating that they are the least robust to uncertainty.

## 6 Conclusion

In summary, the focus of our work is to provide an overview of possible computational approaches for attributing blame in multi-agent sequential decision making. We discuss the strengths and weaknesses of different methods in order to guide practitioners and policy makers in designing tools that support accountability. We conclude that there is no single best choice for blame attribution methods, since there are inherent trade-offs among properties that one might consider important. Looking forward, we recognize several research directions that could address the limitations of our results, some of which we highlight here. a) In this work we primarily focused on the agents' joint return as the outcome of interest. However, it is often important to pinpoint actual causes that led to more fine grained outcomes. Utilizing a causal perspective would be beneficial in this regard and could link our results to prior work (e.g., [16]). b) We considered model-based approaches to blame assignment. Learning blame attribution directly from data (e.g., with model-free counterfactual RL) might be more practical in settings where an approximate model is hard to obtain. c) More generally, ensuring scalability both in the number of agents and the the richness of environments is one of the most important steps for making this work more widely applicable. We deem approaches from multi-agent RL as suitable candidate solutions for resolving this problem. d) We primarily studied blame assignment properties that are taken from or closely relate to those from the game theory literature. This list could be extended and include more principles from moral philosophy and law. For example, in this paper, we adopted a consequentialist approach to blame attribution, focusing on the outcomes of the agents' behavior. Alternatively, one could take a deontological perspective, and focus on the alignment of an agent's behavior with a set of rules. We further discuss different perspectives on blame attribution in the supplementary material. Finally, we would like to draw particular attention to the fact that there is no universal prioritization of properties that applies to all blame attribution problems and hence treating any generic analysis like ours as panacea without further justification, might have a negative impact to the agents that are being blamed. To that end, we would like to emphasize that we see this work not as a final solution to the blame attribution problem, but as a starting point that shows challenges and trade-offs in distributing blame.

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
