# A    List of Appendices

In this section we provide a brief description of the content provided in the appendices of the paper.

- Appendix B provides a table that summarizes the the results in Section 3.
- Appendix C provides additional details on Banzhaf index.
- Appendix D provides additional details on blame attribution under uncertainty.
- Appendix E provides additional details on experimental setup and implementation.
- Appendix F provides an extended discussion on different perspectives on blame attribution and the negative side-effects of under-blaming agents.
- Appendix G contains the proofs of the proposition from Section 3 (Proposition 1, Proposition 2, and Proposition 3).
- Appendix H contains the proof of Theorem 1 from Section 3.
- Appendix I contains the proof of Theorem 2 from Section 3.
- Appendix J contains the proofs of the formal results from Section 4 (Proposition 4, Proposition 5, and Theorem 3).

# B    Table of Methods and Properties

In this section we provide a table that summarizes the results of Section 3 and describes which blame attribution methods satisfy which properties. We use ($\checkmark$) to denote that a method does not satisfy the exact property but a weaker version of it.

|  | $\Psi_{MER}$ | $\Psi_{MC}$ | $\Psi_{SV}$ | $\Psi_{BI}$ | $\Psi_{AP}$ |
|---|---|---|---|---|---|
| $\mathcal{R}_V$ | $\checkmark$ |  | $\checkmark$ |  | $\checkmark$ |
| $\mathcal{R}_E$ |  |  | $\checkmark$ |  | ($\checkmark$) |
| $\mathcal{R}_R$ | $\checkmark$ |  |  |  |  |
| $\mathcal{R}_S$ |  | $\checkmark$ | $\checkmark$ | $\checkmark$ | $\checkmark$ |
| $\mathcal{R}_I$ | $\checkmark$ | $\checkmark$ | $\checkmark$ | $\checkmark$ | $\checkmark$ |
| $\mathcal{R}_{CM}$ |  | $\checkmark$ | $\checkmark$ | $\checkmark$ |  |
| $\mathcal{R}_{PerM}$ |  | $\checkmark$ |  |  | ($\checkmark$) |

Table 1: Summary of the characterization results from Section 3

Method $\Psi_{AP}$ satisfies properties $\mathcal{R}_{AE}$ and $\mathcal{R}_{cPerM}$ which are weaker versions of $\mathcal{R}_E$ and $\mathcal{R}_{PerM}$, respectively.

# C    Banzhaf Index

In this section, we discuss in a greater detail Banzhaf index and its properties. In the context of the sequential decision making setting studied in this paper, Banzhaf Index can be defined as $\beta = \Psi_{BI}(M, \pi^b)$ such that

$$\beta_i = \sum_{S \subseteq \{1,...,n\} \setminus \{i\}} w_S \cdot \left[ J(\pi_{S \cup \{i\}}^{*|\pi^b}, \pi^b_{-S \cup \{i\}}) - J(\pi_S^{*|\pi^b}, \pi^b_{-S}) \right], \qquad (3)$$

where coefficients $w_S$ are set to $w_S = \frac{1}{2^{n-1}}$. The following properties hold:

**Proposition 6.** $\Psi_{BI}(M, \pi^b) = (\beta_1, ..., \beta_n)$, where $\beta_i$ is defined by Eq. (3) and $w_S = \frac{1}{2^{n-1}}$, is a blame attribution method satisfying $\mathcal{R}_S$ (symmetry), $\mathcal{R}_I$ (invariance) and $\mathcal{R}_{CM}$ (contribution monotonicity).

*Proof.* First, notice that Banzhaf Index can be redefined as $\beta = \Psi_{BI}(M, \pi^b)$ such that:

$$\beta_i = \sum_{S \subseteq \{1,...,n\} \setminus \{i\}} w_S \cdot \left[ \Delta_{S \cup \{i\}} - \Delta_S \right]. \qquad (4)$$

We prove the properties as follows:

- $\mathcal{R}_S$ (symmetry): Consider $M$, $\pi^b$, and agents $i$ and $j$ such that $\Delta_{S \cup \{i\}} = \Delta_{S \cup \{j\}}$ for all $S \subseteq \{1, ..., n\} \setminus \{i, j\}$. Notice that $\Delta_{S \cup \{i\}} - \Delta_S = \Delta_{S \cup \{j\}} - \Delta_S$ and $\Delta_{S \cup \{i,j\}} - \Delta_{S \cup \{j\}} = \Delta_{S \cup \{i,j\}} - \Delta_{S \cup \{i\}}$ for all $S \subseteq \{1, ..., n\} \setminus \{i, j\}$. Given the definition of $\beta = \Psi_{BI}(M, \pi^b)$, this implies that $\beta_i = \beta_j$, and hence property $\mathcal{R}_S$ (symmetry) is satisfied.
- $\mathcal{R}_I$ (invariance): Consider $M$, $\pi^b$, and agent $i$ such that $\Delta_{S \cup \{i\}} = \Delta_S$ for all $S$. Given the definition of $\beta = \Psi_{BI}(M, \pi^b)$, this implies that $\beta_i = 0$, and hence property $\mathcal{R}_I$ (invariance) is satisfied.
- $\mathcal{R}_{CM}$ (contribution monotonicity): Consider $M^1$, $\pi^{b^1}$, $M^2$, $\pi^{b^2}$, and agent $i$ such that $\Delta^1_{S \cup \{i\}} - \Delta^1_S \geq \Delta^2_{S \cup \{i\}} - \Delta^2_S$ for all $S$. By using the definitions of $\beta^1 = \Psi_{BI}(M^1, \pi^{b^1})$ and $\beta^2 = \Psi_{BI}(M^2, \pi^{b^2})$, this implies that:

$$\beta_i{}^1 = \sum_{S \subseteq \{1,...,n\} \setminus \{i\}} w_S \cdot \left[ \Delta^1_{S \cup \{i\}} - \Delta^1_S \right] \geq$$

$$\geq \sum_{S \subseteq \{1,...,n\} \setminus \{i\}} w_S \cdot \left[ \Delta^2_{S \cup \{i\}} - \Delta^2_S \right] =$$

$$= \beta_i{}^2,$$

and hence property $\mathcal{R}_{CM}$ (contribution monotonicity) is satisfied. $\qquad \square$

In general, Banzhaf index satisfies a property called 2-efficiency [52] which leads to a slightly different uniqueness result than the one of Theorem 1. This property and the corresponding analysis are out of the scope of this paper, and we refer the reader to [52, 54] for more details.

## D  Additional Information on Blame Attribution under Uncertainty

In this section, we provide additional information on the optimization problems defined in Section 4.1 and the implementation of Blackstone consistent $\widehat{\Psi}_{MER,BC}(M, \mathcal{P}(\pi^b))$ and $\widehat{\Psi}_{AP,BC}(M, \mathcal{P}(\pi^b))$.

### D.1  Implementation of Optimization Problems

In this section, we provide implementation details on the optimization problems defined in Section 4.1, for obtaining Valid and Blackstone consistent blame attribution methods. More specifically, we focus on the optimization problems $\min_{\pi \in \mathcal{P}'(\pi^b)} J(\pi^{*|\pi}_{S \cup \{i\}}, \pi_{-S \cup \{i\}})$ and $\max_{\pi \in \mathcal{P}'(\pi^b)} J(\pi^{*|\pi}_S, \pi_{-S})$, where $S \subseteq \{1, ..., n\}$ and $\mathcal{P}'(\pi^b) \supseteq \mathcal{P}(\pi^b)$. We consider

$$\mathcal{P}(\pi^b) = \Bigg\{ \pi | \pi(a|s) = \pi_1(a_1|s) \cdots \pi_n(a_n|s), \frac{1}{2} \cdot \left\| \pi_i(\cdot|s) - \pi_i^{bas}(\cdot|s) \right\|_1 \leq C, 0 \leq \pi_i(a_i|s) \leq 1,$$

$$\sum_{a_i \in \mathcal{A}_i} \pi_i(a_i|s) = 1 \Bigg\},$$

where $C$ is a non-negative constant and $\pi^{bas}$ is a baseline joint policy. In specific cases, we can set $\mathcal{P}'(\pi^b) = \mathcal{P}(\pi^b)$ and we discuss these cases below. In general, to more directly relate the optimization problems to prior work on robust optimization in MDPs [47, 48], we relax the constraint that $\pi$ factorizes to $\pi(a|s) = \pi_1(a_1|s) \cdots \pi_n(a_n|s)$, and consider

$$\mathcal{P}'(\pi^b) = \Bigg\{ \pi | \prod_{i=1}^n \max(\pi_i^{bas}(a_i|s) - C, 0) \leq \pi(a_1, ..., a_n|s) \leq \prod_{i=1}^n \min(\pi_i^{bas}(a_i|s) + C, 1),$$

$$\sum_{(a_1, ..., a_n) \in \mathcal{A}} \pi(a_1, ..., a_n|s) = 1 \Bigg\}.$$

Notice that since $\sum_{a_i \in \mathcal{A}_i} \pi_i^{bas}(a_i|s) = 1$, we have that $\pi_i^{bas}(a_i|s) - C \leq \pi_i(a_i|s) \leq \pi_i^{bas}(a_i|s) + C$ for every $\pi \in \mathcal{P}(\pi^b)$, and hence $\mathcal{P}''(\pi^b) \supseteq \mathcal{P}(\pi^b)$, where

$$\mathcal{P}''(\pi^b) = \Big\{ \pi | \pi(a|s) = \pi_1(a_1|s) \cdots \pi_n(a_n|s), \max(\pi_i^{bas}(a_i|s) - C, 0) \leq \pi_i(a_i|s)$$

$$\leq \min(\pi_i^{bas}(a_i|s) + C, 1), \sum_{a_i \in \mathcal{A}_i} \pi_i(a_i|s) = 1 \Big\}.$$

Importantly, $\mathcal{P}'(\pi^b) \supseteq \mathcal{P}''(\pi^b)$ implies that $\mathcal{P}'(\pi^b) \supseteq \mathcal{P}(\pi^b)$, which means that $\max_{\pi \in \mathcal{P}'(\pi^b)} J(\pi_S^{*|\tau}, \pi_{-S})$ upper bounds $\max_{\pi \in \mathcal{P}(\pi^b)} J(\pi_S^{*|\tau}, \pi_{-S})$ and $\min_{\pi \in \mathcal{P}'(\pi^b)} J(\pi_{S\cup\{i\}}^{*|\tau}, \pi_{-S\cup\{i\}})$ lower bounds $\min_{\pi \in \mathcal{P}(\pi^b)} J(\pi_{S\cup\{i\}}^{*|\tau}, \pi_{-S\cup\{i\}})$. Therefore, $\min_{\pi \in \mathcal{P}'(\pi^b)} J(\pi_{S\cup\{i\}}^{*|\tau}, \pi_{-S\cup\{i\}})$ and $\max_{\pi \in \mathcal{P}'(\pi^b)} J(\pi_S^{*|\tau}, \pi_{-S})$ can be used for deriving valid and Blackstone consistent blame assignments (e.g., by applying Eq. (1) with the obtained solutions). Next, we discuss how to solve these optimization problems.

While [47, 48] consider uncertainty over transitions dynamics instead of behavior policies, we can solve $\max_{\pi \in \mathcal{P}'(\pi^b)} J(\pi_S^{*|\tau}, \pi_{-S})$ and $\min_{\pi \in \mathcal{P}'(\pi^b)} J(\pi_{S\cup\{i\}}^{*|\tau}, \pi_{-S\cup\{i\}})$ by adapting their robust optimization techniques. To solve the optimization problem $\min_{\pi \in \mathcal{P}'(\pi^b)} J(\pi_{S\cup\{i\}}^{*|\tau}, \pi_{-S\cup\{i\}})$ for subset $S$, we apply the following recursion (in each iteration updating values for each state $s$):

$$\tilde{\pi}(\cdot|s) \leftarrow \underset{\pi(\cdot|s) \in \mathcal{P}'(\pi^b, s)}{\arg\min} \max_{a_{S\cup\{i\}}} \sum_{a_{-S\cup\{i\}}} \pi_{-S\cup\{i\}}(a_{-S\cup\{i\}}|s) \cdot \left[ R(s,a) + \gamma \cdot \sum_{s'} P(s,a,s') \cdot V^k(s') \right],$$

$$V^{k+1}(s) \leftarrow \max_{a_{S\cup\{i\}}} \sum_{a_{-S\cup\{i\}}} \tilde{\pi}_{-S\cup\{i\}}(a_{-S\cup\{i\}}|s) \cdot \left[ R(s,a) + \gamma \cdot \sum_{s'} P(s,a,s') \cdot V^k(s') \right],$$

for $k = 1, 2, \ldots$, where $V : \mathcal{S} \to \mathbb{R}_{\geq 0}$ is the value function, $a_S$ denotes the joint action of agents $S$, $a_{-S}$ denotes the joint action of agents $\{1, ..., n\}\backslash S$, and $a$ is the joint action of all the agents. The optimization problem for finding $\tilde{\pi}$ can be solved via a linear program that minimizes a dummy variable which is constrained to be at least as large as

$$\sum_{a_{-S\cup\{i\}}} \pi_{-S\cup\{i\}}(a_{-S\cup\{i\}}|s) \cdot \left[ R(s,a) + \gamma \cdot \sum_{s'} P(s,a,s') \cdot V^k(s') \right]$$

for all $a_{S\cup\{i\}}$. The optimization problem for finding $V^{k+1}$ can be solved by simply searching over all possible $a_{S\cup\{i\}}$. Similarly, we can solve $\max_{\pi \in \mathcal{P}'(\pi^b)} J(\pi_S^{*|\tau}, \pi_{-S})$ with the following recursion:

$$\tilde{\pi}(\cdot|s) \leftarrow \underset{\pi(\cdot|s) \in \mathcal{P}'(\pi^b, s)}{\arg\max} \max_{a_S} \sum_{a_{-S}} \pi_{-S}(a_{-S}|s) \cdot \left[ R(s,a) + \gamma \cdot \sum_{s'} P(s,a,s') \cdot V^k(s') \right],$$

$$V^{k+1}(s) \leftarrow \max_{a_S} \sum_{a_{-S}} \tilde{\pi}_{-S}(a_{-S}|s) \cdot \left[ R(s,a) + \gamma \cdot \sum_{s'} P(s,a,s') \cdot V^k(s') \right],$$

for $k = 1, 2, \ldots$. The optimization problem for finding $\tilde{\pi}$ can be solved by searching over all $a_S$ and selecting one that maximizes

$$\max_{\pi(\cdot|s) \in \mathcal{P}'(\pi^b, s)} \sum_{a_{-S}} \pi_{-S}(a_{-S}|s) \cdot \left[ R(s,a) + \gamma \cdot \sum_{s'} P(s,a,s') \cdot V^k(s') \right]$$

—the solution to this problem gives us the corresponding $\tilde{\pi}$. The optimization problem for finding $V^{k+1}$ can be solved by searching over all possible $a_S$. The two recursions described above define dynamic programming techniques that are analogs of those in [47, 48], but applied for uncertainty over behavior policies. They can be solved efficiently for smaller action spaces $\mathcal{A}$, e.g., as those in our experiments.

Now, in specific cases, we can set $\mathcal{P}'(\pi^b) = \mathcal{P}(\pi^b)$, which in turn can lead to more efficient blame assignments (since the estimates are tighter). We consider the following two cases:

- First, when there are only two agents in an MMDP, $-S \cup \{i\}$ contains at most one agent. Therefore, we could run the first recursion on $\{\pi_j | \frac{1}{2} \cdot \|\pi_j(\cdot|s) - \pi_j^{bas}(\cdot|s)\|_1 \leq C, 0 \leq \pi_j(a_j|s) \leq 1, \sum_{a_j \in \mathcal{A}_j} \pi_j(a_j|s) = 1\}$ instead of $\{\pi_j | \max(\pi_j^{bas}(a_j|s) - C, 0) \leq \pi_j(a_j|s) \leq \min(\pi_j^{bas}(a_j|s) + C, 1), \sum_{a_j \in \mathcal{A}_j} \pi_j(a_j|s) = 1\}$ and thus solve $\min_{\pi \in \mathcal{P}(\pi^b)} J(\pi_{S\cup\{i\}}^{*|\tau}, \pi_{-S\cup\{i\}})$.

Also in that case, $-S$ contains at most one agent whenever $S \neq \emptyset$, and hence we could run the second recursion on $\{\pi_j | \frac{1}{2} \cdot \|\pi_j(\cdot|s) - \pi_j^{bas}(\cdot|s)\|_1 \leq C, 0 \leq \pi_j(a_j|s) \leq 1, \sum_{a_j \in \mathcal{A}_j} \pi_j(a_j|s) = 1\}$, and thus solve $\max_{\pi \in \mathcal{P}(\pi^b)} J(\pi_S^{*|\pi}, \pi_{-S})$ for every $S \neq \emptyset$. In addition, when the optimal policies of one of the agents, agent $i$, are independent of which policy the other agent, agent $j$, follows we can directly compute an optimal policy for $i$ on $\{\pi_i | \frac{1}{2} \cdot \|\pi_i(\cdot|s) - \pi_i^{bas}(\cdot|s)\|_1 \leq C, 0 \leq \pi_i(a_i|s) \leq 1, \sum_{a_i \in \mathcal{A}_i} \pi_i(a_i|s) = 1\}$, by fixing an arbitrary policy to agent $j$. Then, by fixing agent $i$ to its optimal policy, we can can directly compute an optimal policy of agent $j$ on $\{\pi_j | \frac{1}{2} \cdot \|\pi_j(\cdot|s) - \pi_j^{bas}(\cdot|s)\|_1 \leq C, 0 \leq \pi_j(a_j|s) \leq 1, \sum_{a_j \in \mathcal{A}_j} \pi_j(a_j|s) = 1\}$. This implies that we can run the second recursion directly on $\mathcal{P}(\pi^b)$ for $S = \emptyset$ and thus solve $\max_{\pi \in \mathcal{P}(\pi^b)} J(\pi)$. We use these facts in our experiments for the Gridworld environment, where the optimal policies of $A_1$ are independent of $A_2$'s policy.

- Another specific case is when action spaces $\mathcal{A}_i$ are binary, and in this case, we can directly solve $\max_{\pi \in \mathcal{P}(\pi^b)} J(\pi_S^{*|\pi}, \pi_{-S})$. Namely, we can think of this optimization problem as searching for an optimal joint policy in an MMDP where the actions of agents $-S$ have reduced "influence". Since an optimal joint policy in the reduced MMDP is deterministic, the optimal solution to $\max_{\pi \in \mathcal{P}(\pi^b)} J(\pi_S^{*|\pi}, \pi_{-S})$ sets $\pi_j(a_j|s)$ of agent $j \in -S$ either to its maximum or its minimum value, $\pi_j^{bas}(a_j|s) + C$ and $\pi_j^{bas}(a_j|s) - C$ respectively. In the former case, this means that agent $j$ chooses $a_j$ in the MMDP with the reduced influence, in the latter, this means that agent $j$ chooses the other action. We use this fact in our experiments for the Graph environment.

To conclude, in our experiments we directly solve the optimization problems $\min_{\pi \in \mathcal{P}(\pi^b)} J(\pi_{S \cup \{i\}}^{*|\pi}, \pi_{-S \cup \{i\}})$ and $\max_{\pi \in \mathcal{P}(\pi^b)} J(\pi_S^{*|\pi}, \pi_{-S})$ for the Gridworld environment, and $\max_{\pi \in \mathcal{P}(\pi^b)} J(\pi_S^{*|\pi}, \pi_{-S})$ for the Graph environment.

### D.2 Max-Efficient Rationality and Average Participation under Uncertainty

In this section we discuss the implementation of Blackstone consistent $\widehat{\Psi}_{MER,BC}(M, \mathcal{P}(\pi^b))$ and $\widehat{\Psi}_{AP,BC}(M, \mathcal{P}(\pi^b))$ from Section 4.1. We begin with $\widehat{\Psi}_{MER,BC}(M, \mathcal{P}(\pi^b))$, which can be obtained by solving the optimization problem (P1) with $\Delta_S$ replaced by $\tilde{\Delta}_S = \min_{\pi \in \mathcal{P}'(\pi^b)} J(\pi_S^{*|\pi}, \pi_{-S}) - \max_{\pi \in \mathcal{P}'(\pi^b)} J(\pi)$. A solution to this optimization problem $\widehat{\beta}$ will for at least one solution $\beta$ of (P1) (with $\Delta_S$) satisfy $\widehat{\beta}_i \leq \beta_i$ for all $i$. In that sense, $\widehat{\Psi}_{MER,BC}(M, \mathcal{P}(\pi^b))$ satisfies $\mathcal{R}_{BC}(\Psi_{MER})$ (Blackstone consistency w.r.t. $\Psi_{MER}(M, \pi^b)$). However, note that $\mathcal{R}_{BC}(\Psi_{MER})$ might not hold if (P1) has multiple solutions (e.g., when calculating $\widehat{\Psi}_{MER,BC}$ or $\Psi_{MER}$) and we consider only one solution (e.g., obtained through a tie breaking rule).

Let us now consider $\widehat{\Psi}_{AP,BC}(M, \mathcal{P}(\pi^b))$. $\widehat{\beta} = \widehat{\Psi}_{AP,BC}(M, \mathcal{P}(\pi^b))$ can be implemented as

$$\widehat{\beta}_i = \sum_{S \subseteq \{1,\ldots,n\} \setminus \{i\}} w \cdot \frac{\tilde{c}(M, \mathcal{P}(\pi^b), i)}{|S| + 1} \cdot \tilde{\Delta}_{S \cup \{i\}},$$

where $\tilde{c}(M, \mathcal{P}(\pi^b), i) = \mathbb{1}\left[\widehat{\beta}_{SV,i} > 0\right]$ with $\widehat{\beta}_{SV} = \widehat{\Psi}_{SV,BC}(M, \mathcal{P}(\pi^b))$ (see Section 4.1 for how to calculate $\widehat{\Psi}_{SV,BC}$), $w = \frac{1}{2^n - 1}$ and $\tilde{\Delta}_{S \cup \{i\}} = \min_{\pi \in \mathcal{P}'(\pi^b)} J(\pi_{S \cup \{i\}}^{*|\pi}, \pi_{-S \cup \{i\}}) - \max_{\pi \in \mathcal{P}'(\pi^b)} J(\pi)$. Here, we used the fact that $c$ (in this case, estimate $\tilde{c}$ can be obtained via Shapley value (in this case, Blackstone consistent Shapley value).

## E  Experimental Setup and Implementation Details

In this section, we provide additional information on experimental setup and implementation details.

### E.1  Additional Information on Experimental Setup

**Environment 1:** The exact penalties and rewards of the Gridworld environment (Fig. 1) are as follows: $-0.01$ for blank cells and $S$ cells, $-0.02$ for $F$ cells, $-0.5$ for $H$ cells and $+1$ for cell $G$. Moreover, the cost of intervention $C$ is $-0.05$. The size of the environment's state space is 64

(the state space represents cells of the Gridworld). The action space of agent $A_2$ is $\{0, 1\}$, which corresponds to *don't intervene* and *intervene*, and the action space of agent $A_1$ is $\{0, 1, 2, 3\}$, i.e. *move left*, *move right*, *move up* and *move down*. Note also that the actor remains at the same cell if it takes an action which would take it out of the environment. Finally, the specification of the personal policy of agent $A_1$ can be found in the source code, and more precisely in function `instantiate_behavior_policy_1()` of `env_gridworld.py`.

**Environment 2:** The state space of the Graph environment (Fig. 2) is defined by possible distributions of the 4 agents over the nodes of the graph, 66 states in total. The action space of each agent is $\{0, 1\}$, and the time-horizon of the environment is 5. We test multiple variants of this environment, each of which defines a different reward function. In all the variants, the reward at each time-step $t < 4$ is $+1$ if some formation constraint is satisfied and $-1$ if not, at time-step $t = 4$ the reward is always 0. Next, we describe in more detail the formation constraints and behavior policies for the Graph environment in the first (Coordination) and the second (Robustness) set of experiments.

**Coordination:** In the first set of experiments, we assign weights $w_1 = 1$, $w_2 = 2$, $w_3 = 3$ and $w_4 = 4$ to the four agents. We also consider 4 different formation constraints which are satisfied if $\sum_{i \in \{1,2,3,4\}} w_i \cdot a_i \geq h_m$, where $a_i$ is the action taken by agent $i$ and $h_m$ is a threshold specific to the constraint $m \in \{1, 2, 3, 4\}$. We consider four thresholds: $h_1 = 1$, $h_2 = 7$, $h_3 = 9$ and $h_4 = 10$. For each constraint $m$ to be satisfied, at least $m$ number of agents need to select action 1. Each behavior $\pi^b{}_i$ takes action 0 in every state.

**Robustness:** In the second set of experiments, we consider one formation constraint that is satisfied if agents are arranged equally between the two levels of the graph, $\sum_{i \in \{1,2,3,4\}} a_i = 2$. When the agents are in nodes $-1, 6, 7$ or $8$, each behavior policy $\pi^b{}_i$ takes each action with 0.5 probability. In states where agents are balanced between the levels, each behavior policy $\pi^b{}_i$ takes the action from the previous time-step with probability $p_i$; in unbalanced states, the action that leads to the level with the least number of agents is taken with probability $p_i$. We consider $p_i = 1 - (i - 1) \cdot 0.2$ for each agent $i \in \{1, 2, 3, 4\}$.

Discount factor $\gamma$ is set to 0.99 in both environments.

### E.2 Implementation Details

The solutions to the evaluation and optimization problems utilized by the blame attribution methods can be computed efficiently using standard (robust) optimization techniques. In the source code, the solvers of these problems are implemented as the following functions:

- policy performance evaluation in function `recursion_1_a()` (see `recursion_graph.py` and `recursion_gridworld.py`).
- problem $\arg\max_{\pi_S} J(\pi_S, \pi^b{}_{-S})$ in functions `recursion_1_c()` (see `recursion_graph.py`) and `recursion_1_c_ag1()`, `recursion_1_c_ag2()` (see `recursion_gridworld.py`).
- problem $\arg\max_{\pi \in \mathcal{P}'(\pi^b)} J(\pi)$ in function `recursion_2_a()` (see `recursion_graph.py` and `recursion_gridworld.py`).
- problem $\min_{\pi \in \mathcal{P}'(\pi^b)} J(\pi^{*|\pi}_{S \cup \{i\}}, \pi_{-S \cup \{i\}})$ in functions `recursion_3_a()` (see `recursion_graph.py`) and `recursion_3_a_ag1()`, `recursion_3_a_ag2()` (see `recursion_gridworld.py`).
- problem $\max_{\pi \in \mathcal{P}'(\pi^b)} J(\pi^{*|\pi}_S, \pi_{-S})$ in functions `recursion_3_b()` (see `recursion_graph.py`) and `recursion_3_b_ag1()`, `recursion_3_b_ag2()` (see `recursion_gridworld.py`).

### E.3 Solutions to the Optimization Problem (P1)

(P1) might have multiple optimal solutions. Therefore, when calculating $\Psi_{MER}$ (Section 3.1) or $\widehat{\Psi}_{MER,BC}$ (Section 4.1 and Appendix D), a way to decide which solution is going to be the blame assignment output is needed. For the experiments on the Gridworld environment the optimal solution assigning the maximum blame to $A_2$ was always selected. For the experiments on the Graph environment, an LP solver was applied: in the case of the Graph environment, our experiments only require the total blame assigned to the agents so any optimal solution to the LP produces the same results (see below).

$L_1$ **Distance:** For the Max-Efficient Rationality method in Fig. 3c and 3g of Section 5, we consider the $L_1$ distance between an output $\widehat{\beta}$ of the consistent method $\widehat{\Psi}_{MER,BC}$ and an output $\beta$ of $\Psi_{MER}$, such that $\widehat{\beta}_i \leq \beta_i$ for all $i$. Notice that the $L_1$ distance between any two such blame assignments is equal to their difference in total blame, $\sum_{i \in \{1,\dots,n\}} |\beta_i - \widehat{\beta}_i| = \sum_{i \in \{1,\dots,n\}} \beta_i - \sum_{i \in \{1,\dots,n\}} \widehat{\beta}_i$. Notice also that the total blame $\sum_{i \in \{1,\dots,n\}} \beta_i$ (resp. $\sum_{i \in \{1,\dots,n\}} \widehat{\beta}_i$) is the same for all optimal solutions $\beta$ (resp. $\widehat{\beta}$) of (P1) with $\Delta_S$ (resp. $\widetilde{\Delta}_S$), since they maximize the same objective. Hence, for obtaining the $L_1$ distance between the output of the consistent method $\widehat{\Psi}_{MER,BC}$ and its "targeted assignment", it suffices to compute the difference $\sum_{i \in \{1,\dots,n\}} \beta_i - \sum_{i \in \{1,\dots,n\}} \widehat{\beta}_i$ for any two optimal solutions $\beta$ and $\widehat{\beta}$.

**Total Blame:** The total blame assigned by the Max-Efficient Rationality method in each of the figures 3e, 3d and 3h of Section 5 remains the same for all the optimal solutions of (P1).

### E.4  Total Amount of Compute and Type of Resources

All experiments were run on a personal laptop (with Intel Core i7-8750H CPU). Experiments were also run multiple times for 10 different seeds, and we report averages and standard deviations. The total running time of the experiments on the Gridworld environment is a few minutes ($\sim$10) and of the experiments on the Graph environment a few hours ($\sim$3). Tables 2 and 3 show how much (CPU) time it takes to compute Shapley value under uncertainty (using the approaches from Section 4), for $\epsilon_{max} = \{0.01, 0.05, 0.1, 0.15, 0.2\}$. Note that $\Psi_{SV}$ does not depend on $\epsilon_{max}$—its running time for the Gridworld environment is $0.453125 \pm 0.19111$ sec and for the Graph environment is $2.02187 \pm 0.06853$ sec.

|  | $\widehat{\Psi}_{SV}$ | $\widehat{\Psi}_{SV,V}$ | $\widehat{\Psi}_{SV,BC}$ |
|---|---|---|---|
| $\epsilon_{max} = 0.05$ | $0.45625 \pm 0.19848$ | $1.19843 \pm 0.51044$ | $1.38906 \pm 0.60939$ |
| $\epsilon_{max} = 0.10$ | $0.46093 \pm 0.20049$ | $1.21093 \pm 0.55609$ | $1.45781 \pm 0.71592$ |
| $\epsilon_{max} = 0.15$ | $0.47187 \pm 0.20925$ | $1.14062 \pm 0.51864$ | $1.45468 \pm 0.66985$ |
| $\epsilon_{max} = 0.20$ | $0.46093 \pm 0.22011$ | $1.20937 \pm 0.60934$ | $1.72500 \pm 0.96822$ |

Table 2: Running times of different approaches for SV under uncertainty on the Gridworld environment. All times are measured in seconds (sec).

|  | $\widehat{\Psi}_{SV}$ | $\widehat{\Psi}_{SV,V}$ | $\widehat{\Psi}_{SV,BC}$ |
|---|---|---|---|
| $\epsilon_{max} = 0.01$ | $2.06250 \pm 0.10892$ | $3.80625 \pm 0.13243$ | $92.38750 \pm 1.59190$ |
| $\epsilon_{max} = 0.05$ | $2.07031 \pm 0.18077$ | $3.91718 \pm 0.18944$ | $91.60000 \pm 1.43320$ |
| $\epsilon_{max} = 0.10$ | $1.97500 \pm 0.05466$ | $3.84218 \pm 0.12798$ | $92.84375 \pm 3.45815$ |

Table 3: Running times of different approaches for SV under uncertainty on the Graph environment. All times are measured in seconds (sec).

$\widehat{\Psi}_{SV,BC}$ has the largest computing time, while $\Psi_{SV}$ and $\widehat{\Psi}_{SV}$ have the lowest computing times. These results are not surprising given that $\Psi_{SV}$ and $\widehat{\Psi}_{SV}$ only need to compute the values once and they are not running robust optimization. Moreover, $\widehat{\Psi}_{SV,BC}$ solves $\min_{\pi \in \mathcal{P}'(\pi^b)} J(\pi_{S\cup\{i\}}^{*|\pi}, \pi_{-S\cup\{i\}})$ and $\max_{\pi \in \mathcal{P}'(\pi^b)} J(\pi_S^{*|\pi}, \pi_{-S})$ for each $S$ separately, unlike $\widehat{\Psi}_{SV,V}$, which only requires robust

optimization for finding a solution to the optimization problem $\arg\max_{\pi \in \mathcal{P}'(\pi^b)} J(\pi)$. The running times of methods that compute $\widehat{\Psi}_{SV}$ do not appear to have strong dependency on $\epsilon_{max}$. This is expected for $\widehat{\Psi}_{SV}$ since it is based on point estimates, and does not use robust optimization.

Note that the computation results obtained when calculating the aforementioned Shapley value blame assignments can be reused in computing the blame assignments of the other blame attribution methods, which we do in our experiments.

## F    Extended Discussion

This section of the appendix discusses different perspectives on blame attribution, and the potential negative side-effects of under-blaming agents.

### F.1    Different Perspectives on Blame Attribution

**Consequentialism:** In this paper we follow a *consequentialist* [55] approach to the blame attribution problem, in the sense that we consider the amount of an agent's blame to depend solely on the outcome of its policy. More specifically, we consider blame attribution methods and desirable properties that measure how good or bad an agent's policy is based only on the inefficiency it causes to the multi-agent system.[9] A common objection to this type of approaches is that they do not blame an agent for violating common ground rules, i.e. they concentrate only on the ends rather than the means [56]. For example, consider an intersection accident scenario that involves two drivers: the first driver, $D_1$, proceeds north and the second driver $D_2$ proceeds east, both of them drive below the speed limit. Assume that $D_2$ violates a stop sign but could not do anything different to avoid the accident, while if $D_1$ would drive above the speed limit then with high probability the accident would have been avoided. According to consequentialism, in this example driver $D_1$ deserves more blame than $D_2$, although $D_2$ is the one that breaks the law.

**Deontology:** Consequentialism is often contrasted to another major approach in normative ethics, *deontology* [55, 57]. From a deontological perspective, the quality of an agent's policy is based on how well it follows a clear set of rules or duties[10], rather than its consequences. Therefore, a deontological approach to blame attribution would assign more blame to the second driver, from the example above, because they violate a well-known traffic regulation. Of course, deontological approaches face criticism too, for instance people argue that deontological ethics are rigid—they focus on rules, ignoring the (potentially) severe consequences of one's behavior [58]. For instance, avoiding a car crash may be more important than not violating the speed limit in the example above.

The problem of assigning blame is inherently multi-dimensional and can be viewed through both deontological and consequentialist lenses (among others). In this paper we take a consequentialist viewpoint because it provides clear and practical guidance, at least when estimating (counterfactual) outcomes is plausible. However, we do not see the two normative ethical theories as mutually exclusive [59], and thus our intention is not to replace deontological approaches, but to complement them.

### F.2    Under-Blaming Agents

Apart from serving justice, blame attribution is also important for incentivizing decision makers to adopt policies that will minimize the system's inefficiency. To that end, we introduce in Section 2.3 the performance monotonicity property, the purpose of which is to motivate agents to individually improve their policies. The second property we introduce, Blackstone consistency, aims to ensure that no agent will be over-blamed when the behavior policies are not fully known to the blame attribution procedure. As expected, experimental results from Section 5 show that Blackstone consistent methods end up under-blaming agents instead. Just like over-blaming, under-blaming has its own adverse effects. Such an effect is incentivizing bad behaviors, since the agents receive reduced penalties. Therefore, there seems to be a trade-off between ensuring that no one is unjustly blamed under uncertainty and providing incentives for good behavior.

---

[9]This is well-aligned with the main idea of *utilitarianism* [55], which measures how good or bad an action is based only on the overall utility of its consequences.

[10]Deontology takes root from the Greek word *deon*, which means duty.

# G Proofs of the Propositions from Section 3

This section of the appendix contains the proofs of the propositions from Section 3, in particular: Proposition 1, Proposition 2, and Proposition 3.

## G.1 Proof of Proposition 1

**Proposition 1.** *Every solution to the optimization problem* (P1), *i.e.,* $\Psi_{MER}$, *satisfies* $\mathcal{R}_V$ *(validity),* $\mathcal{R}_R$ *(rationality) and* $\mathcal{R}_I$ *(invariance).*

*Proof.* We prove the properties as follows:

- $\mathcal{R}_V$ (validity): Consider $M$, $\pi^b$. Every solution to the optimization problem (P1), i.e., $\beta = \Psi_{MER}(M, \pi^b)$, satisfies the constraint $\sum_{i \in \{1,...,n\}} \beta_i \leq \Delta_{\{1,...,n\}}$. The last inequality can be rewritten as $\sum_{i=1}^n \beta_i \leq \Delta$, and hence property $\mathcal{R}_V$ (validity) is satisfied.
- $\mathcal{R}_R$ (rationality): Consider $M$, $\pi^b$ and $S \subseteq \{1,...,n\}$. Every solution to the optimization problem (P1), i.e., $\beta = \Psi_{MER}(M, \pi^b)$, satisfies the constraint $\sum_{i \in S} \beta_i \leq \Delta_S$, and hence property $\mathcal{R}_R$ (rationality) is satisfied.
- $\mathcal{R}_I$ (invariance): Consider $M$, $\pi^b$, and an agent $i$ such that $\Delta_{S \cup \{i\}} = \Delta_S$ for all $S$. This implies that $\Delta_i = \Delta_\emptyset = 0$. Now, due to the constraints of the optimization problem (P1), every solution to the optimization problem (P1), i.e., $\beta = \Psi_{MER}(M, \pi^b)$, satisfies the constraint $\beta_i \leq \Delta_i = 0$. Note also that $\sum_{j \in S} \beta_j \leq \Delta_{S \cup \{i\}} - \beta_i = \Delta_S - \beta_i$, but also $\sum_{j \in S} \beta_j \leq \Delta_S$ (where $i \notin S$). Therefore, the constraints in which agent $i$ participates can be replaced by the the constraint $\beta_i \leq 0$. Together with the fact that the objective function is the total blame, this implies that the optimal $\beta_i$ is independent of $\beta_j$ ($j \neq i$), and furthermore that its value is equal to $\beta_i = 0$. Hence, property $\mathcal{R}_I$ (invariance) is satisfied.

$\square$

## G.2 Proof of Proposition 2

**Proposition 2.** $\Psi_{MC}(M, \pi^b) = (\Delta_1, ..., \Delta_n)$ *satisfies* $\mathcal{R}_S$ *(symmetry),* $\mathcal{R}_I$ *(invariance),* $\mathcal{R}_{CM}$ *(contribution monotonicity) and* $\mathcal{R}_{PerM}$ *(performance monotonicity).*

*Proof.* We prove the properties as follows:

- $\mathcal{R}_S$ (symmetry): Consider $M$, $\pi^b$, and agents $i$ and $j$ such that $\Delta_{S \cup \{i\}} = \Delta_{S \cup \{j\}}$ for all $S \subseteq \{1,...,n\} \setminus \{i,j\}$. Notice that $\Delta_i = \Delta_j$. By using the definition of $\beta = \Psi_{MC}(M, \pi^b)$, we have that $\beta_i = \Delta_i = \Delta_j = \beta_j$. Hence, property $\mathcal{R}_S$ (symmetry) is satisfied.
- $\mathcal{R}_I$ (invariance): Consider $M$, $\pi^b$, and agent $i$ such that $\Delta_{S \cup \{i\}} = \Delta_S$ for all $S$. Given the definition of $\beta = \Psi_{MC}(M, \pi^b)$, this implies that $\beta_i = \Delta_i = \Delta_\emptyset = 0$. Hence, property $\mathcal{R}_I$ (invariance) is satisfied.
- $\mathcal{R}_{CM}$ (contribution monotonicity): Consider $M^1, \pi^{b^1}, M^2, \pi^{b^2}$, and agent $i$ such that $\Delta^1_{S \cup \{i\}} - \Delta^1_S \geq \Delta^2_{S \cup \{i\}} - \Delta^2_S$ for all $S$. By using the definitions of $\beta^1 = \Psi_{MC}(M^1, \pi^{b^1})$ and $\beta^2 = \Psi_{MC}(M^2, \pi^{b^2})$, we have that $\beta_i^1 = \Delta_i^1 = \Delta^1_{\emptyset \cup \{i\}} - \Delta^1_\emptyset \geq \Delta^2_{\emptyset \cup \{i\}} - \Delta^2_\emptyset = \Delta_i^2 = \beta_i^2$. Hence, property $\mathcal{R}_{CM}$ (contribution monotonicity) is satisfied.
- $\mathcal{R}_{PerM}$ (performance monotonicity): Consider $M$, $\pi^b{}_{-i}$, $\pi_i$ and $\pi'_i$ such that $J(\pi_i, \pi^b{}_{-i}) \leq J(\pi'_i, \pi^b{}_{-i})$. This implies that:

$$J(\pi_i, \pi^b{}_{-i}) \leq J(\pi'_i, \pi^b{}_{-i}) \Rightarrow$$
$$\Rightarrow J(\pi_i^{*|\pi^b}, \pi^b{}_{-i}) - J(\pi_i, \pi^b{}_{-i}) \geq J(\pi_i^{*|\pi^b}, \pi^b{}_{-i}) - J(\pi'_i, \pi^b{}_{-i}) \Rightarrow$$
$$\Rightarrow \Delta_i \geq \Delta'_i.$$

By using the definitions of $\beta = \Psi_{MC}(M, (\pi_i, \pi^b{}_{-i}))$ and $\beta' = \Psi_{MC}(M, (\pi'_i, \pi^b{}_{-i}))$, we obtain that $\beta_i = \Delta_i \geq \Delta'_i = \beta_i'$. Hence, property $\mathcal{R}_{PerM}$ (performance monotonicity) is satisfied.

$\square$

### G.3 Proof of Proposition 3

**Proposition 3.** *No blame attribution method $\Psi$ satisfies $\mathcal{R}_E$ (efficiency), $\mathcal{R}_S$ (symmetry), $\mathcal{R}_I$ (invariance) and $\mathcal{R}_{PerM}$ (performance monotonicity).*

*Proof.* We prove the stated impossibility result by contradiction. Suppose that there is a blame attribution method $\Psi$ that satisfies $\mathcal{R}_E$ (efficiency), $\mathcal{R}_S$ (symmetry), $\mathcal{R}_I$ (invariance) and $\mathcal{R}_{PerM}$ (performance monotonicity).

Consider an MMDP $M$ with two agents $\{1, 2\}$, two states—the initial state and the terminal state—and the action space $\mathcal{A} = \{0, 1, 2\} \times \{0, 1, 2\}$. In the initial state, the agents obtain zero reward when they both take action $0$, reward equal to $2$ when one of them takes action $0$ and the other one action $2$ or they both take action $2$, and reward equal to $0.9$ when they take any other pair of actions. After the agents perform their actions in the initial state, the MMDP transitions to the terminal state. Consider also the deterministic policies: $\pi^b{}_2$ that takes action $0$, $\pi_1$ that takes action $0$ and $\pi'_1$ that takes action $1$, in the initial state.

We have the following three observations:

- Note that $J(\pi_1, \pi^b{}_2) \leq J(\pi'_1, \pi^b{}_2)$ and hence from property $\mathcal{R}_{PerM}$ (performance monotonicity) we have that $\beta_1 \geq \beta'_1$, where $\beta = \Psi(M, (\pi_1, \pi^b{}_2))$ and $\beta' = \Psi(M, (\pi'_1, \pi^b{}_2))$.
- Note that $\Delta_{\{1\}} = \Delta_{\{2\}} = 2$ and thus from property $\mathcal{R}_S$ (symmetry) it follows that $\beta_1 = \beta_2$. Also, from property $\mathcal{R}_E$ (efficiency) we have that $\beta_1 + \beta_2 = \Delta = 2$, and hence $\beta_1 = 1$ and $\beta_2 = 1$.
- Note that $\Delta'_{\{2\}} = 0$ and $\Delta'_{\{1,2\}} = \Delta'_{\{1\}} = 1.1$ and thus from property $\mathcal{R}_I$ (invariance) it follows that $\beta'_2 = 0$. From property $\mathcal{R}_E$ (efficiency) we have that $\beta'_1 + \beta'_2 = \Delta' = 1.1$ and hence $\beta'_1 = 1.1$, which contradicts the first two observations.

$\square$

## H   Proof of Theorem 1

In this section, we provide a proof of Theorem 1. Since this proof utilizes the results of [60], we first provide some background details on these results.

### H.1   Background

To prove the uniqueness result for the Shapley Value method, Theorem 1, we use a result from [60]. Before we embark on the proof, we set the necessary background. Let $N$ be a set, such that $N \neq \emptyset$, $|N| < \infty$, and $u : 2^N \to \mathbb{R}$ be a function such that $u(\emptyset) = 0$. Then we call $N$ set of agents and $u$ game, and denote with $\mathcal{G}^N$ the class of games with player set $N$. We say that a game $u \in \mathcal{G}^N$ is *monotone*, if for each $S, T \subseteq N$, $S \subseteq T$; $u(S) \leq u(T)$. Moreover, we say that function $\psi : G \to \mathbb{R}^N$ is a solution on the class $G \in \mathcal{G}^N$. Next, we state three axioms from [60]:

- *Pareto Optimality (PO)*: We say that a solution $\psi$ on class of games $G \subseteq \mathcal{G}^N$ satisfies *PO* (Pareto optimality), if for each game $u \in G$: $\sum_{i \in N} \psi_i(u) = u(N)$.
- *Equal Treatment Property (ETP)*: We say that a solution $\psi$ on class of games $G \subseteq \mathcal{G}^N$ satisfies *ETP* (equal treatment property), if for each game $u \in G$ and $i, j \in N$; $\psi_i(u) = \psi_j(u)$, whenever $u(S \cup \{i\}) - u(S) = u(S \cup \{j\}) - u(S)$ for every $S \subseteq N \setminus \{i, j\}$.
- *Marginality (M)*: We say that a solution $\psi$ on class of games $G \subseteq \mathcal{G}^N$ satisfies *M* (marginality), if for all games $u, v \in G$ and $i \in N$: $\psi_i(u) = \psi_i(v)$, whenever $u(S \cup \{i\}) - u(S) = v(S \cup \{i\}) - v(S)$ for every $S \subseteq N$.

We also define the Shapley value method for this setting. For any game $u \in \mathcal{G}^N$, the Shapley value solution $\phi$ is given by

$$\phi_i(u) = \sum_{S \subseteq N \setminus \{i\}} w_S \cdot [u(S \cup \{i\}) - u(S)], \tag{5}$$

where coefficients $w_S$ are set to $w_S = \frac{|S|!(|N|-|S|-1)!}{|N|!}$.

Next we restate Theorem 3.9 from [60]:

**Theorem 4.** *Solution $\psi$ defined on the class of monotone games satisfies axiom PO (Pareto optimality), ETP (equal treatment Property) and M (marginality), iff it is the Shapley value solution.*

We introduce a slightly different axiom than *M* (marginality):

- *Unequal Marginality (UM)*: We say that a solution $\psi$ on class of games $G \subseteq \mathcal{G}^N$ satisfies *UM* (unequal marginality), if for all games $u, v \in G$ and $i \in N$: $\psi_i(u) \geq \psi_i(v)$, whenever $u(S \cup \{i\}) - u(S) \geq v(S \cup \{i\}) - v(S)$ for every $S \subseteq N$.

We also state a Corollary of Theorem 4:

**Corollary 1.** *Solution $\psi$ defined on the class of monotone games satisfies axiom PO (Pareto optimality), ETP (equal treatment Property) and UM (unequal marginality), iff it is the Shapley value solution.*

*Proof.* We prove that Shapley value solution $\phi$ satisfies axiom *UM* (unequal marginality). Consider monotone games $u, v$, and agent $i \in N$ such that $u(S \cup \{i\}) - u(S) \geq v(S \cup \{i\}) - v(S)$ for every $S \subseteq N$, then;

$$\phi_i(u) = \sum_{S \subseteq N \setminus \{i\}} w_S \cdot [u(S \cup \{i\}) - u(S)] \geq$$

$$\geq \sum_{S \subseteq N \setminus \{i\}} w_S \cdot [v(S \cup \{i\}) - v(S)] =$$

$$= \phi_i(v).$$

Since *UM* (unequal marginality) is a stronger axiom than *M* (marginality), and Shapley value solution satisfies it, the uniqueness result stated in the Corollary holds because of Theorem 4. $\square$

Consider $M$, $\pi^b$ and notice that $\Delta_\emptyset = J(\pi^b) - J(\pi^b) = 0$. We say that set of agents $N$ and game $u$ are defined by $M$, $\pi^b$, if $N = \{1, ..., n\}$ and $u(S) = \Delta_S$ for every $S$. We denote with $\mathcal{H}$ the class of games that can be defined in that way. Let $\psi_{SV}$ be the solution on class $\mathcal{H}$ such that for every $M$, $\pi^b$, $\psi_{SV}(u) = \Psi_{SV}(M, \pi^b)$, where game $u$ is defined by $M$, $\pi^b$. Given Eq. (5), this implies that $\psi_{SV}$ is the Shapley Value solution on $\mathcal{H}$.

We state three simple lemmas that show a one to one correspondence between the axioms *PO* (Pareto optimality), *ETP* (equal treatment property) and *UM* (unequal marginality) and blame attribution properties:

**Lemma 1.** *Let $\Psi$ be a blame attribution method and $\psi$ a solution on $\mathcal{H}$, such that for every $M$, $\pi^b$, $\Psi(M, \pi^b) = \psi(u)$, where game $u$ is defined by $M$, $\pi^b$. Then, $\Psi$ satisfies $\mathcal{R}_E$ (efficiency) iff $\psi$ satisfies PO (Pareto optimality) on $\mathcal{H}$.*

*Proof.* Consider $M$, $\pi^b$ and game $u$ defined by $M$, $\pi^b$. Then the statement is true because $u(N) = \Delta_{\{1,...,n\}} = \Delta$. $\square$

**Lemma 2.** *Let $\Psi$ be a blame attribution method and $\psi$ a solution on $\mathcal{H}$, such that for every $M$, $\pi^b$, $\Psi(M, \pi^b) = \psi(u)$, where game $u$ is defined by $M$, $\pi^b$. Then, $\Psi$ satisfies $\mathcal{R}_S$ (symmetry) iff $\psi$ satisfies ETP (equal treatment property) on $\mathcal{H}$.*

*Proof.* Consider $M$, $\pi^b$ and game $u$ defined by $M$, $\pi^b$. Given that $u(S) = \Delta_S$ for every $S$, we have that for every $i$ and $j$, $\Delta_{S \cup \{i\}} - \Delta_S = \Delta_{S \cup \{j\}} - \Delta_S$ iff $u(S \cup \{i\}) - u(S) = u(S \cup \{j\}) - u(S)$. Hence, the statement is true. $\square$

**Lemma 3.** *Let $\Psi$ be a blame attribution method and $\psi$ a solution on $\mathcal{H}$, such that for every $M$, $\pi^b$, $\Psi(M, \pi^b) = \psi(u)$, where game $u$ is defined by $M$, $\pi^b$. Then, $\Psi$ satisfies $\mathcal{R}_{CM}$ (contribution monotonicity) iff $\psi$ satisfies UM (unequal marginality) on $\mathcal{H}$.*

*Proof.* Consider $M^1$, $\pi^{b1}$ and $M^2$, $\pi^{b2}$, and games $u^1$ and $u^2$ defined by $M^1$, $\pi^{b1}$ and $M^2$, $\pi^{b2}$, respectively. Given that $u^1(S) = \Delta_S^1$ and $u^2(S) = \Delta_S^2$ for every $S$, we have that for every $i$, $\Delta_{S \cup \{i\}}^1 - \Delta_S^1 \geq \Delta_{S \cup \{i\}}^2 - \Delta_S^2$ iff $u^1(S \cup \{i\}) - u^1(S) \geq u^2(S \cup \{i\}) - u^2(S)$. Hence, the statement is true. $\square$

## H.2 Proof

**Theorem 1.** $\Psi_{SV}(M, \pi^b) = (\beta_1, ..., \beta_n)$, *where $\beta_i$ is defined by Eq.* (1) *and* $w_S = \frac{|S|!(n-|S|-1)!}{n!}$, *is a unique blame attribution method satisfying $\mathcal{R}_E$ (efficiency), $\mathcal{R}_S$ (symmetry) and $\mathcal{R}_{CM}$ (contribution monotonicity). Additionally, $\Psi_{SV}$ satisfies $\mathcal{R}_V$ (validity) and $\mathcal{R}_I$ (invariance).*

*Proof.* Consider $M, \pi^b$ and game $u$ defined by $M, \pi^b$. Consider also $S$ and $T$ such that $S \subseteq T$. We have that:

$$J(\pi_T^{*|\pi^b}, \pi^b{}_{-T}) \geq J(\pi_S^{*|\pi^b}, \pi^b{}_{-S}) \Rightarrow$$
$$\Rightarrow J(\pi_T^{*|\pi^b}, \pi^b{}_{-T}) - J(\pi^b) \geq J(\pi_S^{*|\pi^b}, \pi^b{}_{-S}) - J(\pi^b) \Rightarrow$$
$$\Rightarrow \Delta_T \geq \Delta_S \Rightarrow u(T) \geq u(S).$$

This implies that class $\mathcal{H}$ consists only of monotone games, and hence by Corollary 1 we have that $\psi_{SV}$ is a unique solution on $\mathcal{H}$ satisfying *PO* (Pareto optimality), *ETP* (equal treatment property) and *UM* (unequal marginality). Given Lemmas 1, 2, and 3, this implies that $\Psi_{SV}$ is a unique blame attribution method satisfying $\mathcal{R}_E$ (efficiency), $\mathcal{R}_S$ (symmetry) and $\mathcal{R}_{CM}$ (contribution monotonicity).

We also prove the properties $\mathcal{R}_V$ (validity) and $\mathcal{R}_I$ (invariance) as follows:

- $\mathcal{R}_V$ (validity): Consider $M, \pi^b$. Given that $\Psi_{SV}$ satisfies property $\mathcal{R}_E$ (efficiency), it holds that $\sum_{i \in \{1,...,n\}} \beta_i = \Delta$. Hence, property $\mathcal{R}_V$ (validity) is satisfied.
- $\mathcal{R}_I$ (invariance): Consider $M, \pi^b$, and agent $i$ such that $\Delta_{S \cup \{i\}} = \Delta_S$ for all $S$. This implies that $J(\pi_{S \cup \{i\}}^{*|\pi^b}, \pi^b{}_{-S \cup \{i\}}) = J(\pi_S^{*|\pi^b}, \pi^b{}_{-S})$ for all $S$. Given the definition of $\Psi_{SV}(M, \pi^b)$, we have that $\beta_i = 0$. Hence, property $\mathcal{R}_I$ (invariance) is satisfied.

$\square$

# I   Proof of Theorem 2

Before we proceed with the proof of Theorem 2, notice that the contribution function $c$ from Section 3.4 can be rewritten in the equivalent form:

$$c(M, \pi^b, i) = \begin{cases} 0 & \text{if } \Delta_{S \cup \{i\}} = \Delta_S, \quad \forall S \subseteq \{1, ..., n\} \\ 1 & \text{otherwise} \end{cases}.$$

We also state the following lemmas:

**Lemma 4.** *Consider a function $f : 2^{\{1,...,n\}} \to \mathbb{R}_{\geq 0}$. There exist some MMDP $M$ and agents' behavior joint policy $\pi^b$ such that the marginal inefficiency of every subset of agents $S$ is equal to $f(S)$, iff $f(\emptyset) = 0$ and $f(S_1) \leq f(S_2)$ whenever $S_1 \subseteq S_2$, where $S_1$ and $S_2$ are subsets of $\{1, ..., n\}$.*

*Proof.* First, we show that the conditions on function $f$ are necessary:

- Suppose that there exist $M, \pi^b$ such that $\Delta_{\emptyset} > 0$. Given the definition of marginal inefficiency this would imply that $J(\pi^b) > J(\pi^b)$. Hence, we reach a contradiction.
- Suppose that there exist $M, \pi^b$ such that $\Delta_{S_1} > \Delta_{S_2}$, where $S1 \subseteq S_2$. Given the definition of marginal inefficiency this would imply that $J(\pi_{S_1}^{*|\pi^b}, \pi^b{}_{-S_1}) > J(\pi_{S_2}^{*|\pi^b}, \pi^b{}_{-S_2})$. Hence, we reach a contradiction.

Next we show that the conditions on function $f$ are sufficient. Consider an MMDP $M$ with two states—the initial state and the terminal state—and the action space $\mathcal{A} = \times_{i=1}^{n} \{0, 1\}$. In the initial state, the agents obtain zero reward when they all take action 0 and reward $f(S)$ when agents in $S$ take action 1 and the rest of the agents take action 0. Consider also the deterministic joint policy $\pi^b$, where every agent takes action 0. Notice that $J(\pi^b) = 0$.

For every subset of agents $S$ it holds that $J(\pi_S^{*|\pi^b}, \pi^b{}_{-S}) = f(S)$, because taking action 1 is the best that every agent in $S$ can do. Hence, for the marginal inefficiency of $S$ we have that $\Delta_S = J(\pi_S^{*|\pi^b}, \pi^b{}_{-S}) - J(\pi^b) = f(S)$.

$\square$

**Lemma 5.** *Let $\Psi$ satisfy $\mathcal{R}_{cParM}$ (c-participation monotonicity). Then, for every $M^1$, $\pi^{b^1}$ and $M^2$, $\pi^{b^2}$ such that $c(M^1, \pi^{b^1}, i) = c(M^2, \pi^{b^2}, i)$ for every $i$, $\beta_i^{\,1} = \beta_i^{\,2}$ whenever $\Delta^1_{S\cup\{i\}} = \Delta^2_{S\cup\{i\}}$ for all $S$, where $\beta^1 = \Psi(M^1, \pi^{b^1})$ and $\beta^2 = \Psi(M^2, \pi^{b^2})$.*

*Proof.* Consider agent $i$ such that $\Delta^1_{S\cup\{i\}} = \Delta^2_{S\cup\{i\}}$ for all $S$. Given that $\Psi$ satisfies $\mathcal{R}_{cParM}$ (c-participation monotonicity), this implies $\beta_i^{\,1} \geq \beta_i^{\,2}$ and $\beta_i^{\,1} \leq \beta_i^{\,2}$, and hence $\beta_i^{\,1} = \beta_i^{\,2}$. $\square$

**Lemma 6.** *Let $\Psi$ satisfy $\mathcal{R}_{RcParM}$ (relative c-participation monotonicity). Then, for every $M^1$, $\pi^{b^1}$ and $M^2$, $\pi^{b^2}$ such that $c(M^1, \pi^{b^1}, i) = c(M^2, \pi^{b^2}, i)$ for every $i$, $\beta_j^{\,1} - \beta_j^{\,2} = \beta_k^{\,1} - \beta_k^{\,2}$ whenever $c(M^1, \pi^{b^1}, j) = c(M^1, \pi^{b^1}, k)$ and $\Delta^1_{S\cup\{j\}} - \Delta^2_{S\cup\{j\}} = \Delta^1_{S\cup\{k\}} - \Delta^2_{S\cup\{k\}}$ for every $S \subseteq \{1, ..., n\}\backslash\{j, k\}$, where $\beta^1 = \Psi(M^1, \pi^{b^1})$ and $\beta^2 = \Psi(M^2, \pi^{b^2})$.*

*Proof.* Consider agents $j$ and $k$ such that $c(M^1, \pi^{b^1}, j) = c(M^1, \pi^{b^1}, k)$ and $\Delta^1_{S\cup\{j\}} - \Delta^2_{S\cup\{j\}} = \Delta^1_{S\cup\{k\}} - \Delta^2_{S\cup\{k\}}$ for all $S \subseteq \{1, ..., n\}\backslash\{j, k\}$. Given that $\Psi$ satisfies $\mathcal{R}_{RcParM}$ (relative c-participation monotonicity), this implies $\beta_j^{\,1} - \beta_j^{\,2} \geq \beta_k^{\,1} - \beta_k^{\,2}$ and $\beta_j^{\,1} - \beta_j^{\,2} \leq \beta_k^{\,1} - \beta_k^{\,2}$, and hence $\beta_j^{\,1} - \beta_j^{\,2} = \beta_k^{\,1} - \beta_k^{\,2}$. $\square$

### Proof of Theorem 2

**Theorem 2.** $\Psi_{AP}(M, \pi^b) = (\beta_1, ..., \beta_n)$, where $\beta_i$ is defined by Eq. (2) and $w = \frac{1}{2^n - 1}$, is a unique blame attribution method that satisfies $\mathcal{R}_{AE}$ (average-efficiency), $\mathcal{R}_S$ (symmetry), $\mathcal{R}_I$ (invariance), $\mathcal{R}_{cParM}$ (c-participation monotonicity) and $\mathcal{R}_{RcParM}$ (relative c-participation monotonicity). Furthermore, $\Psi_{AP}$ satisfies $\mathcal{R}_{cPerM}$ (c-performance monotonicity) and $\mathcal{R}_V$ (validity).

*Proof.* The proof is separated into two parts. In the first part we prove that $\Psi_{AP}$ satisfies the mentioned properties, while in the second part we show that if a blame attribution method satisfies all mentioned properties, it must be the $\Psi_{AP}$ method.

#### First Part

We prove the properties as follows:

- $\mathcal{R}_{AE}$ (average-efficiency): Consider $M, \pi^b$. By using the definition of $\beta = \Psi_{AP}(M, \pi^b)$:

$$
\begin{aligned}
\sum_{i=1}^{n} \beta_i &= \sum_{i=1}^{n} \sum_{S \subseteq \{1,...,n\}\backslash\{i\}} w \cdot \frac{c(M, \pi^b, i)}{\sum_{j \in S} c(M, \pi^b, j) + 1} \cdot \Delta_{S\cup\{i\}} = \\
&= \frac{1}{2^n - 1} \cdot \sum_{i \in \{1,...,n\}\,|\,c(M,\pi^b,i)=1} \sum_{S \subseteq \{1,...,n\}\backslash\{i\}} \frac{1}{\sum_{j \in S} c(M, \pi^b, j) + 1} \cdot \Delta_{S\cup\{i\}} = \\
&= \frac{1}{2^n - 1} \cdot \sum_{i \in \{1,...,n\}\,|\,c(M,\pi^b,i)=1} \sum_{S \subseteq \{1,...,n\}\,|\,i \in S} \frac{1}{\sum_{j \in S} c(M, \pi^b, j)} \cdot \Delta_S = \\
&= \frac{1}{2^n - 1} \cdot \sum_{S \subseteq \{1,...,n\}} \sum_{i \in S\,|\,c(M,\pi^b,i)=1} \frac{1}{\sum_{j \in S} c(M, \pi^b, j)} \cdot \Delta_S = \\
&= \frac{1}{2^n - 1} \cdot \sum_{S \subseteq \{1,...,n\}} \Delta_S,
\end{aligned}
$$

  and hence property $\mathcal{R}_{AE}$ (average-efficiency) is satisfied.
- $\mathcal{R}_S$ (symmetry): Consider $M, \pi^b$, and agents $i$ and $j$ such that $\Delta_{S\cup\{i\}} = \Delta_{S\cup\{j\}}$ for all $S \subseteq \{1, ..., n\}\backslash\{i, j\}$. Notice that if $\Delta_{S\cup\{i\}} = \Delta_S$ for all $S$ then $\Delta_{S\cup\{j\}} = \Delta_S$ and $\Delta_{S\cup\{i,j\}} = \Delta_{S\cup\{j\}} = \Delta_{S\cup\{i\}}$ for every $S \subseteq \{1, ..., n\}\backslash\{i, j\}$, and hence $\Delta_{S\cup\{j\}} = \Delta_S$ for all $S$. Given the definition of contribution function $c$, this implies that if $c(M, \pi^b, i) = 0$, then $c(M, \pi^b, j) = 0$.

For similar reasons, it also holds that if $c(M, \pi^b, j) = 0$, then $c(M, \pi^b, i) = 0$, and hence $c(M, \pi^b, i) = c(M, \pi^b, j)$. By using the definition of $\beta = \Psi_{AP}(M, \pi^b)$, we have that:

$$\beta_i = \sum_{S \subseteq \{1,...,n\} \setminus \{i\}} w \cdot \frac{c(M, \pi^b, i)}{\sum_{k \in S} c(M, \pi^b, k) + 1} \cdot \Delta_{S \cup \{i\}} =$$

$$= \sum_{S \subseteq \{1,...,n\} \setminus \{i,j\}} w \cdot \frac{c(M, \pi^b, i)}{\sum_{k \in S} c(M, \pi^b, k) + 1} \cdot \Delta_{S \cup \{i\}} +$$

$$+ \sum_{S \subseteq \{1,...,n\} \setminus \{i,j\}} w \cdot \frac{c(M, \pi^b, i)}{\sum_{k \in S \cup \{j\}} c(M, \pi^b, k) + 1} \cdot \Delta_{S \cup \{i,j\}} =$$

$$= \sum_{S \subseteq \{1,...,n\} \setminus \{i,j\}} w \cdot \frac{c(M, \pi^b, j)}{\sum_{k \in S} c(M, \pi^b, k) + 1} \cdot \Delta_{S \cup \{j\}} +$$

$$+ \sum_{S \subseteq \{1,...,n\} \setminus \{i,j\}} w \cdot \frac{c(M, \pi^b, j)}{\sum_{k \in S \cup \{i\}} c(M, \pi^b, k) + 1} \cdot \Delta_{S \cup \{i,j\}} =$$

$$= \sum_{S \subseteq \{1,...,n\} \setminus \{j\}} w \cdot \frac{c(M, \pi^b, j)}{\sum_{k \in S} c(M, \pi^b, k) + 1} \cdot \Delta_{S \cup \{j\}} = \beta_j,$$

and hence property $\mathcal{R}_S$ (symmetry) is satisfied.

- $\mathcal{R}_I$ (invariance): Consider $M$, $\pi^b$, and agent $i$ such that $\Delta_{S \cup \{i\}} = \Delta_S$ for all $S$. Given the definitions of contribution function $c$ and $\beta = \Psi_{AP}(M, \pi^b)$, this implies that $\beta_i = 0$. Hence, property $\mathcal{R}_I$ (invariance) is satisfied.

- $\mathcal{R}_{cParM}$ (c-participation monotonicity): Consider $M^1$, $\pi^{b^1}$ and $M^2$, $\pi^{b^2}$ such that $c(M^1, \pi^{b^1}, i) = c(M^2, \pi^{b^2}, i)$ for every $i$. Consider also agent $i$ such that $\Delta^1_{S \cup \{i\}} \geq \Delta^2_{S \cup \{i\}}$ for all $S$. By using the definitions of $\beta^1 = \Psi_{AP}(M^1, \pi^{b^1})$ and $\beta^2 = \Psi_{AP}(M^2, \pi^{b^2})$, this implies:

$$\beta_i^1 = \sum_{S \subseteq \{1,...,n\} \setminus \{i\}} w \cdot \frac{c(M^1, \pi^{b^1}, i)}{\sum_{j \in S} c(M^1, \pi^{b^1}, j) + 1} \cdot \Delta^1_{S \cup \{i\}} =$$

$$= \sum_{S \subseteq \{1,...,n\} \setminus \{i\}} w \cdot \frac{c(M^2, \pi^{b^2}, i)}{\sum_{j \in S} c(M^2, \pi^{b^2}, j) + 1} \cdot \Delta^1_{S \cup \{i\}} \geq$$

$$\geq \sum_{S \subseteq \{1,...,n\} \setminus \{i\}} w \cdot \frac{c(M^2, \pi^{b^2}, i)}{\sum_{j \in S} c(M^2, \pi^{b^2}, j) + 1} \cdot \Delta^2_{S \cup \{i\}} = \beta_i^2,$$

and hence property $\mathcal{R}_{cParM}$ (c-participation monotonicity) is satisfied.

- $\mathcal{R}_{RcParM}$ (relative c-participation monotonicity): Consider $M^1$, $\pi^{b^1}$ and $M^2$, $\pi^{b^2}$ such that $c(M^1, \pi^{b^1}, i) = c(M^2, \pi^{b^2}, i)$ for every $i$. Consider also agents $j$ and $k$ such that $c(M^1, \pi^{b^1}, j) = c(M^1, \pi^{b^1}, k)$ and $\Delta^1_{S \cup \{j\}} - \Delta^2_{S \cup \{j\}} \geq \Delta^1_{S \cup \{k\}} - \Delta^2_{S \cup \{k\}}$ for all $S \subseteq \{1,...,n\} \setminus \{j,k\}$. By using the definitions of $\beta^1 = \Psi_{AP}(M^1, \pi^{b^1})$ and $\beta^2 = \Psi_{AP}(M^2, \pi^{b^2})$, this implies:

$$\beta_j^1 - \beta_j^2 = \sum_{S \subseteq \{1,...,n\} \setminus \{j\}} w \cdot \frac{c(M^1, \pi^{b^1}, j)}{\sum_{i \in S} c(M^1, \pi^{b^1}, i) + 1} \cdot \left[ \Delta^1_{S \cup \{j\}} - \Delta^2_{S \cup \{j\}} \right] =$$

$$= \sum_{S \subseteq \{1,...,n\} \setminus \{j,k\}} w \cdot \frac{c(M^1, \pi^{b^1}, j)}{\sum_{i \in S} c(M^1, \pi^{b^1}, i) + 1} \cdot \left[ \Delta^1_{S \cup \{j\}} - \Delta^2_{S \cup \{j\}} \right] +$$

$$+ \sum_{S \subseteq \{1,...,n\} \setminus \{j,k\}} w \cdot \frac{c(M^1, \pi^{b^1}, j)}{\sum_{i \in S \cup \{k\}} c(M^1, \pi^{b^1}, i) + 1} \cdot \left[ \Delta^1_{S \cup \{j,k\}} - \Delta^2_{S \cup \{j,k\}} \right] \geq$$

$$\geq \sum_{S \subseteq \{1,...,n\} \setminus \{j,k\}} w \cdot \frac{c(M^1, \pi^{b^1}, k)}{\sum_{i \in S} c(M^1, \pi^{b^1}, i) + 1} \cdot \left[ \Delta^1_{S \cup \{k\}} - \Delta^2_{S \cup \{k\}} \right] +$$

$$+ \sum_{S\subseteq\{1,...,n\}\setminus\{j,k\}} w \cdot \frac{c(M^1,\pi^{b^1},k)}{\sum_{i\in S\cup\{j\}} c(M^1,\pi^{b^1},i)+1} \cdot \left[\Delta^1_{S\cup\{j,k\}} - \Delta^2_{S\cup\{j,k\}}\right] =$$

$$= \sum_{S\subseteq\{1,...,n\}\setminus\{k\}} w \cdot \frac{c(M^1,\pi^{b^1},k)}{\sum_{i\in S} c(M^1,\pi^{b^1},i)+1} \cdot \left[\Delta^1_{S\cup\{k\}} - \Delta^2_{S\cup\{k\}}\right] = \beta_k{}^1 - \beta_k{}^2,$$

and hence property $\mathcal{R}_{RcParM}$ (relative c-participation monotonicity) is satisfied.

- $\mathcal{R}_{cPerM}$ (c-performance monotonicity): Consider $M$, $\pi^b{}_{-i}$, $\pi_i$ and $\pi'_i$ such that $J(\pi_i,\pi^b{}_{-i}) \leq J(\pi'_i,\pi^b{}_{-i})$ and $c(M,(\pi_i,\pi^b{}_{-i}),j) = c(M,(\pi'_i,\pi^b{}_{-i}),j)$ for every $j$. This implies that:

$$J(\pi_i,\pi^b{}_{-i}) \leq J(\pi'_i,\pi^b{}_{-i}) \Rightarrow$$
$$\Rightarrow J(\pi^{*|\pi^b}_{S\cup\{i\}},\pi^b{}_{-S\cup\{i\}}) - J(\pi_i,\pi^b{}_{-i}) \geq J(\pi^{*|\pi^b}_{S\cup\{i\}},\pi^b{}_{-S\cup\{i\}}) - J(\pi'_i,\pi^b{}_{-i}) \Rightarrow$$
$$\Rightarrow \Delta_{S\cup\{i\}} \geq \Delta'_{S\cup\{i\}}$$

for every $S \subseteq \{1,...,n\}\setminus\{i\}$. Given the definitions of $\beta = \Psi_{AP}(M,(\pi_i,\pi^b{}_{-i}))$ and $\beta' = \Psi_{AP}(M,(\pi'_i,\pi^b{}_{-i}))$, this implies that $\beta_i \geq \beta_i{}'$. Hence, property $\mathcal{R}_{cPerM}$ (c-performance monotonicity) is satisfied.

- $\mathcal{R}_V$ (validity): Consider $M$, $\pi^b$. Notice that $\sum_{S\subseteq\{1,...,n\}} \frac{1}{2^n-1} \cdot \Delta_S \leq \Delta$. Given that $\Psi_{AP}$ satisfies property $\mathcal{R}_{AE}$ (average efficiency), we have that $\sum_{i=1}^n \beta_i = \sum_{S\subseteq\{1,...,n\}} \frac{1}{2^n-1} \cdot \Delta_S$, and thus $\sum_{i=1}^n \beta_i \leq \Delta$, where $\beta = \Psi_{AP}(M,\pi^b)$. Hence property $\mathcal{R}_V$ (validity) is satisfied.

**Second Part**

We begin by introducing some additional notation. Consider $M$, $\pi^b$. We define the sets of agents $C_0 = \{i \in \{1,\ldots,n\} : c(M,\pi^b,i) = 0\}$ and $C_1 = \{i \in \{1,\ldots,n\} : c(M,\pi^b,i) = 1\}$. Consider $M^\epsilon$, $\pi^{b^\epsilon}$ such that:

$$\Delta_S^\epsilon = \begin{cases} \Delta_S + \epsilon & \text{if } S \cap C_1 \neq \emptyset \\ \Delta_S & \text{otherwise,} \end{cases} \tag{6}$$

where $\epsilon > 0$. Note that for every subset $S$ such that $S \cap C_1 = \emptyset$ it holds that $\Delta_S^\epsilon = 0$, but we use $\Delta_S^\epsilon = \Delta_S$ for notational simplicity. Moreover, notice that Eq. (6) satisfies the conditions of Lemma 4, and hence $M^\epsilon$, $\pi^{b^\epsilon}$ exist.

We prove that $\Psi_{AP}$ uniquely satisfies the properties mentioned in Theorem 2 through two intermediate lemmas. Lemma 7 states that if $\Psi(M^\epsilon,\pi^{b^\epsilon}) = \Psi_{AP}(M^\epsilon,\pi^{b^\epsilon})$ then $\Psi(M,\pi^b) = \Psi_{AP}(M,\pi^b)$, and Lemma 8 states that $\Psi(M^\epsilon,\pi^{b^\epsilon}) = \Psi_{AP}(M^\epsilon,\pi^{b^\epsilon})$.

**Lemma 7.** *Consider $M$, $\pi^b$ and $M^\epsilon$, $\pi^{b^\epsilon}$, where $\Delta_S^\epsilon$ is defined by Eq. (6). If $\Psi$ satisfies properties $\mathcal{R}_{AE}$, $\mathcal{R}_S$, $\mathcal{R}_I$, $\mathcal{R}_{cParM}$ and $\mathcal{R}_{RcParM}$ and $\Psi(M^\epsilon,\pi^{b^\epsilon}) = \Psi_{AP}(M^\epsilon,\pi^{b^\epsilon})$, then $\Psi(M,\pi^b) = \Psi_{AP}(M,\pi^b)$.*

*Proof.* We state three claims that we prove after the end of the proof of Theorem 2:

**Claim 1.** $c(M,\pi^b,i) = c(M^\epsilon,\pi^{b^\epsilon},i)$ *for every* $i$.

**Claim 2.** $\beta_i = 0$ *and* $\beta_i{}^\epsilon = 0$ *for every* $i \in C_0$, *where* $\beta = \Psi(M,\pi^b)$ *and* $\beta^\epsilon = \Psi(M^\epsilon,\pi^{b^\epsilon})$.

**Claim 3.** $\beta_i{}^\epsilon - \beta_i = r$ *for every* $i \in C_1$, *where* $\beta = \Psi(M,\pi^b)$ *and* $\beta^\epsilon = \Psi(M^\epsilon,\pi^{b^\epsilon})$, *and* $r = \frac{1}{|C_1|} \cdot \sum_{S\subseteq\{1,...,n\}} w \cdot [\Delta_S^\epsilon - \Delta_S]$.

Given Claim 3, the assumption $\Psi(M^\epsilon,\pi^{b^\epsilon}) = \Psi_{AP}(M^\epsilon,\pi^{b^\epsilon})$ implies that for every $i \in C_1$:

$$\beta_i = \sum_{S\subseteq\{1,...,n\}\setminus\{i\}} w \cdot \frac{1}{\sum_{j\in S} c(M^\epsilon,\pi^{b^\epsilon},j)+1} \cdot \Delta_{S\cup\{i\}}^\epsilon - \frac{1}{|C_1|} \cdot \sum_{S\subseteq\{1,...,n\}} w \cdot [\Delta_S^\epsilon - \Delta_S].$$
$$\tag{7}$$

Combining Claim 2 and Eq. (7) implies that:

$$\beta_i = c(M, \pi^b, i) \cdot \left[ \sum_{S \subseteq \{1,...,n\}\setminus\{i\}} w \cdot \frac{1}{\sum_{j \in S} c(M^\epsilon, \pi^{b^\epsilon}, j) + 1} \cdot \Delta^\epsilon_{S \cup \{i\}} - \frac{1}{|C_1|} \cdot \sum_{S \subseteq \{1,...,n\}} w \cdot [\Delta^\epsilon_S - \Delta_S] \right].$$

(8)

Notice that $\beta = \Psi(M, \pi^b)$ is uniquely defined by the properties of $\Psi$, Eq. (8), and since $\Psi_{AP}$ satisfies all properties assumed for $\Psi$ (see Part 1), it must hold that $\Psi(M, \pi^b) = \Psi_{AP}(M, \pi^b)$. This concludes the proof of Lemma 7. $\qquad\square$

**Lemma 8.** *Consider $M$, $\pi^b$ and $M^\epsilon$, $\pi^{b^\epsilon}$, where $\Delta^\epsilon_S$ is defined by Eq. (6). If $\Psi$ satisfies properties $\mathcal{R}_{AE}$, $\mathcal{R}_S$, $\mathcal{R}_I$, $\mathcal{R}_{cParM}$ and $\mathcal{R}_{RcParM}$, then $\Psi(M^\epsilon, \pi^{b^\epsilon}) = \Psi_{AP}(M^\epsilon, \pi^{b^\epsilon})$.*

*Proof.* Let $I = \{1, ..., 2^{|C_1|} - 1\}$ be an index set, and for each $\iota \in I$, let $S_\iota$ be a subset of $C_1$ other than $\emptyset$. We assume that the indexing of subsets $S \subseteq C_1$ satisfies the following condition: for every $\iota, \zeta \in I$, $\iota < \zeta$ whenever $|S_\iota| > |S_\zeta|$.

Consider $M$, $\pi^b$ and $M^\epsilon$, $\pi^{b^\epsilon}$, where $\Delta^\epsilon_S$ is defined by Eq. (6). For each index number $\iota \in I$ consider $M^\iota$, $\pi^{b^\iota}$ such that:

$$\Delta^\iota_S = \begin{cases} \epsilon & \text{if } S \cap C_1 = S_\zeta, \text{ where } \zeta > \iota \\ \Delta_S & \text{if } S \cap C_1 = \emptyset \\ \Delta_S + \epsilon & \text{otherwise,} \end{cases}$$

(9)

where $\epsilon > 0$. Note that for every subset $S$ such that $S \cap C_1 = \emptyset$ it holds that $\Delta^\iota_S = 0$, but we use $\Delta^\iota_S = \Delta_S$ for notational simplicity. Moreover, notice that for every $\iota \in I$ Eq. (9) satisfies the conditions of Lemma 4, and hence $M^\iota$, $\pi^{b^\iota}$ exist. Notice also that $\Delta^{2^{|C_1|}-1}_S = \Delta^\epsilon_S$ for every $S$.

We state four claims that we prove after the end of the proof of Theorem 2:

**Claim 4.** *For each $\iota \in I$, $c(M, \pi^b, i) = c(M^\iota, \pi^{b^\iota}, i)$ for every $i$.*

**Claim 5.** *For each $\iota \in I$, $\beta_i^\iota = 0$ for every $i \in C_0$, where $\beta^\iota = \Psi(M^\iota, \pi^{b^\iota})$.*

**Claim 6.** *For each $\iota \in I \setminus \{2^{|C_1|} - 1\}$, $\beta_i^{\iota+1} - \beta_i^\iota = 0$ for every $i \in C_1 \setminus S_{\iota+1}$, where $\beta^\iota = \Psi(M^\iota, \pi^{b^\iota})$ and $\beta^{\iota+1} = \Psi(M^{\iota+1}, \pi^{b^{\iota+1}})$.*

**Claim 7.** *For each $\iota \in I \setminus \{2^{|C_1|} - 1\}$, $\beta_i^{\iota+1} - \beta_i^\iota = r$ for every $i \in S_{\iota+1}$, where $\beta^\iota = \Psi(M^\iota, \pi^{b^\iota})$ and $\beta^{\iota+1} = \Psi(M^{\iota+1}, \pi^{b^{\iota+1}})$, and $r = \frac{1}{|S_{\iota+1}|} \cdot \sum_{S \subseteq \{1,...,n\}} w \cdot [\Delta^{\iota+1}_S - \Delta^\iota_S]$.*

We prove that $\Psi(M^{2^{|C_1|}-1}, \pi^{b^{2^{|C_1|}-1}}) = \Psi_{AP}(M^{2^{|C_1|}-1}, \pi^{b^{2^{|C_1|}-1}})$, by using induction in the index number $\iota$. Note that because $\Delta^{2^{|C_1|}-1}_S = \Delta^\epsilon_S$ for every $S$, showing $\Psi(M^{2^{|C_1|}-1}, \pi^{b^{2^{|C_1|}-1}}) = \Psi_{AP}(M^{2^{|C_1|}-1}, \pi^{b^{2^{|C_1|}-1}})$ is equivalent to showing that $\Psi(M^\epsilon, \pi^{b^\epsilon}) = \Psi_{AP}(M^\epsilon, \pi^{b^\epsilon})$.

$\iota = 1$: We show that $\Psi(M^1, \pi^{b^1}) = \Psi_{AP}(M^1, \pi^{b^1})$. Because of the condition that the indexing of the subsets of $C_1$ has to satisfy, it follows that $S_1 = C_1$. Notice that for every two agents $i, j \in C_1$ it holds that $S \cup \{i\} \cap C_1 \neq C_1 = S_1$ and $S \cup \{j\} \cap C_1 \neq C_1 = S_1$, for every $S \subseteq \{1, ..., n\} \setminus \{i, j\}$. By using Eq. (9), this implies that $\Delta^1_{S \cup \{i\}} = \epsilon = \Delta^1_{S \cup \{j\}}$, for every $S \subseteq \{1, ..., n\} \setminus \{i, j\}$. Given that $\Psi$ is assumed to satisfy $\mathcal{R}_S$ (symmetry), this implies that $\beta_i^1 = \beta_j^1$, where $\beta^1 = \Psi(M^1, \pi^{b^1})$. It follows that for every $i \in C_1$:

$$\beta_i^1 = \frac{1}{|C_1|} \cdot \sum_{j \in C_1} \beta_j^1.$$

By using Claim 5, we have that $\beta_i^1 = \frac{1}{|C_1|} \cdot \sum_{j \in \{1,...,n\}} \beta_j^1$. Given that $\Psi$ satisfies $\mathcal{R}_{AE}$ (average efficiency), this implies that for every $i \in C_1$:

$$\beta_i^1 = \frac{1}{|C_1|} \cdot \sum_{j \in \{1,...,n\}} \beta_j^1 = \frac{1}{|C_1|} \cdot \sum_{S \subseteq \{1,...,n\}} w \cdot \Delta^1_S.$$

(10)

Combining Claim 5 and Eq. (10) implies that:

$$\beta_i{}^1 = c(M, \pi^b, i) \cdot \frac{1}{|C_1|} \cdot \sum_{S \subseteq \{1,\dots,n\}} w \cdot \Delta_S^1. \tag{11}$$

Notice that $\beta^1 = \Psi(M^1, \pi^{b^1})$ is uniquely defined by the properties of $\Psi$, Eq. (11), and since $\Psi_{AP}$ satisfies all properties assumed for $\Psi$ (see Part 1), it must hold that $\Psi_{AP}(M^1, \pi^{b^1}) = \beta^1$, and hence $\Psi(M^1, \pi^{b^1}) = \Psi_{AP}(M^1, \pi^{b^1})$.

$\iota \in I \backslash \{2^{|C_1|} - 1\}$: Given that $\Psi(M^\iota, \pi^{b^\iota}) = \Psi_{AP}(M^\iota, \pi^{b^\iota})$, we show that $\Psi(M^{\iota+1}, \pi^{b^{\iota+1}}) = \Psi_{AP}(M^{\iota+1}, \pi^{b^{\iota+1}})$.

By using the definition of $\Psi_{AP}(M^\iota, \pi^{b^\iota})$ and Claim 6, the assumption $\Psi(M^\iota, \pi^{b^\iota}) = \Psi_{AP}(M^\iota, \pi^{b^\iota})$ implies that for every $i \in C_1 \backslash S_{\iota+1}$:

$$\beta_i{}^{\iota+1} = \beta_i{}^\iota = \sum_{S \subseteq \{1,\dots,n\} \backslash \{i\}} w \cdot \frac{1}{\sum_{j \in S} c(M^\iota, \pi^{b^\iota}, j) + 1} \cdot \Delta_{S \cup \{i\}}^\iota. \tag{12}$$

By using the definition of $\Psi_{AP}(M^\iota, \pi^{b^\iota})$ and Claim 7, the assumption $\Psi(M^\iota, \pi^{b^\iota}) = \Psi_{AP}(M^\iota, \pi^{b^\iota})$ implies that for every $i \in S_{\iota+1}$:

$$\begin{aligned}
\beta_i{}^{\iota+1} = &\beta_i{}^\iota + \frac{1}{|S_{\iota+1}|} \cdot \sum_{S \subseteq \{1,\dots,n\}} w \cdot \left[ \Delta_S^{\iota+1} - \Delta_S^\iota \right] = \\
= &\sum_{S \subseteq \{1,\dots,n\} \backslash \{i\}} w \cdot \frac{1}{\sum_{j \in S} c(M^\iota, \pi^{b^\iota}, j) + 1} \cdot \Delta_{S \cup \{i\}}^\iota + \frac{1}{|S_{\iota+1}|} \cdot \sum_{S \subseteq \{1,\dots,n\}} w \cdot \left[ \Delta_S^{\iota+1} - \Delta_S^\iota \right].
\end{aligned} \tag{13}$$

Notice that $\beta^{\iota+1} = \Psi(M^{\iota+1}, \pi^{b^{\iota+1}})$ is uniquely defined by properties of $\Psi$, Claim 5, Eq. (12) and Eq. (13), and since $\Psi_{AP}$ satisfies all the properties assumed for $\Psi$ (see Part 1), it must hold that $\Psi(M^{\iota+1}, \pi^{b^{\iota+1}}) = \beta^{\iota+1}$, and hence $\Psi(M^{\iota+1}, \pi^{b^{\iota+1}}) = \Psi_{AP}(M^{\iota+1}, \pi^{b^{\iota+1}})$. This concludes the induction step and the proof of Lemma 8. $\qquad \square$

The second part of the proof is hence concluded. $\qquad \square$

**Proofs of the Claims 1, 2 and 3**

**Claim 1.** $c(M, \pi^b, i) = c(M^\epsilon, \pi^{b^\epsilon}, i)$ *for every $i$.*

*Proof.* Consider agent $i$ such that $i \in C_1$. Given Eq. (6), this implies that $\Delta_i^\epsilon = \Delta_i + \epsilon > 0 = \Delta_\emptyset^\epsilon$, and thus $c(M^\epsilon, \pi^{b^\epsilon}, i) = 1$. Hence, $c(M, \pi^b, i) = c(M^\epsilon, \pi^{b^\epsilon}, i)$.

Consider agent $i$ such that $i \in C_0$. Given the definition of contribution function $c$, we have that $\Delta_{S \cup \{i\}} = \Delta_S$ for every $S$. By using Eq. (6), this implies that $\Delta_{S \cup \{i\}}^\epsilon = \Delta_S^\epsilon$ for every $S$ such that $S \cap C_1 = \emptyset$ and $\Delta_{S \cup \{i\}}^\epsilon = \Delta_{S \cup \{i\}} + \epsilon = \Delta_S + \epsilon = \Delta_S^\epsilon$ for every $S$ such that $S \cap C_1 \neq \emptyset$, and thus $c(M^\epsilon, \pi^{b^\epsilon}, i) = 0$. Hence, $c(M, \pi^b, i) = c(M^\epsilon, \pi^{b^\epsilon}, i)$. $\qquad \square$

**Claim 2.** $\beta_i = 0$ *and* $\beta_i^\epsilon = 0$ *for every $i \in C_0$, where $\beta = \Psi(M, \pi^b)$ and $\beta^\epsilon = \Psi(M^\epsilon, \pi^{b^\epsilon})$.*

*Proof.* Given the definition of contribution function $c$, the lemma follows from Claim 1 and the assumption that $\Psi$ satisfies property $\mathcal{R}_I$ (invariance). $\qquad \square$

**Claim 3.** $\beta_i^\epsilon - \beta_i = r$ *for every $i \in C_1$, where $\beta = \Psi(M, \pi^b)$ and $\beta^\epsilon = \Psi(M^\epsilon, \pi^{b^\epsilon})$, and $r = \frac{1}{|C_1|} \cdot \sum_{S \subseteq \{1,\dots,n\}} w \cdot [\Delta_S^\epsilon - \Delta_S]$.*

*Proof.* Notice that for every two agents $j, k \in C_1$ it holds that $c(M, \pi^b, j) = c(M, \pi^b, k)$ and that $\Delta_{S \cup \{j\}}^\epsilon - \Delta_{S \cup \{j\}} = \Delta_{S \cup \{k\}}^\epsilon - \Delta_{S \cup \{k\}} = \epsilon$ for every $S \subseteq \{1,\dots,n\} \backslash \{j,k\}$. Furthermore,

from Claim 1 we have that $c(M, \pi^b, i) = c(M^\epsilon, \pi^{b^\epsilon}, i)$ for every $i$, and thus Lemma 6 applies, $\beta_j{}^\epsilon - \beta_j = \beta_k{}^\epsilon - \beta_k = r$, where $r$ is some constant. Notice that:

$$r = \frac{1}{|C_1|} \cdot \sum_{i \in C_1} \beta_i{}^\epsilon - \beta_i.$$

By using Claim 2, we have that $r = \frac{1}{|C_1|} \cdot \sum_{i \in \{1,...,n\}} \beta_i{}^\epsilon - \beta_i$. Given that $\Psi$ is assumed to satisfy $\mathcal{R}_{AE}$ (average efficiency), this implies that:

$$r = \frac{1}{|C_1|} \cdot \sum_{i \in \{1,...,n\}} \beta_i{}^\epsilon - \beta_i = \frac{1}{|C_1|} \cdot \sum_{S \subseteq \{1,...,n\}} w \cdot [\Delta_S^\epsilon - \Delta_S],$$

and hence $\beta_i{}^\epsilon - \beta_i = \frac{1}{|C_1|} \cdot \sum_{S \subseteq \{1,...,n\}} w \cdot [\Delta_S^\epsilon - \Delta_S]$ for every $i \in C_1$. $\qquad \square$

**Proofs of the Claims 4, 5, 6 and 7**

**Claim 4.** *For each $\iota \in I$, $c(M, \pi^b, i) = c(M^\iota, \pi^{b^\iota}, i)$ for every $i$.*

*Proof.* Consider agent $i$ such that $i \in C_1$. Given Eq. (9), this implies that $\Delta_i^\iota \geq \epsilon > 0 = \Delta_\emptyset^\iota$, and thus $c(M^\iota, \pi^{b^\iota}, i) = 1$. Hence, $c(M, \pi^b, i) = c(M^\iota, \pi^{b^\iota}, i)$.

Consider agent $i$ such that $i \in C_0$. Given the definition of contribution function $c$, we have that $\Delta_{S \cup \{i\}} = \Delta_S$ for every $S$. By using Eq. (9), this implies that $\Delta_{S \cup \{i\}}^\iota = \epsilon = \Delta_S^\iota$ for every $S$ such that $S \cap C_1 = S_\zeta$, where $\zeta > \iota$, $\Delta_{S \cup \{i\}}^\iota = \Delta_S^\iota$ for every $S$ such that $S \cap C_1 = \emptyset$ and $\Delta_{S \cup \{i\}}^\iota = \Delta_{S \cup \{i\}} + \epsilon = \Delta_S + \epsilon = \Delta_S^\iota$ for every other $S$, and thus $c(M^\iota, \pi^{b^\iota}, i) = 0$. Hence, $c(M, \pi^b, i) = c(M^\iota, \pi^{b^\iota}, i)$. $\qquad \square$

**Claim 5.** *For each $\iota \in I$, $\beta_i^\iota = 0$ for every $i \in C_0$, where $\beta^\iota = \Psi(M^\iota, \pi^{b^\iota})$.*

*Proof.* Given the definition of contribution function $c$, the lemma follows from Claim 4 and the assumption that $\Psi$ satisfies property $\mathcal{R}_I$ (invariance). $\qquad \square$

Based on the next observation we prove the rest of the claims:

**Observation 1.** *Observe that for each $\iota \in I \setminus \{2^{|C_1|} - 1\}$, $\Delta_S^{\iota+1} = \Delta_S^\iota$ for every $S$ such that $S \cap C_1 \neq S_{\iota+1}$.*[11]

**Claim 6.** *For each $\iota \in I \setminus \{2^{|C_1|} - 1\}$, $\beta_i^{\iota+1} - \beta_i^\iota = 0$ for every $i \in C_1 \setminus S_{\iota+1}$, where $\beta^\iota = \Psi(M^\iota, \pi^{b^\iota})$ and $\beta^{\iota+1} = \Psi(M^{\iota+1}, \pi^{b^{\iota+1}})$.*

*Proof.* Notice that for every agent $i \in C_1 \setminus S_{\iota+1}$ it holds that $S \cup \{i\} \cap C_1 \neq S_{\iota+1}$ for every $S$. Given Observation 1 this implies that $\Delta_{S \cup \{i\}}^{\iota+1} = \Delta_{S \cup \{i\}}^\iota$ for every $S$. Furthermore, from Claim 4 we have that $c(M, \pi^b, i) = c(M^\iota, \pi^{b^\iota}, i) = c(M^{\iota+1}, \pi^{b^{\iota+1}}, i)$ for every $i$, and thus Lemma 5 applies, and for every $i \in C_1 \setminus S_{\iota+1}$ we have that $\beta_i^{\iota+1} = \beta_i^\iota$. $\qquad \square$

**Claim 7.** *For each $\iota \in I \setminus \{2^{|C_1|} - 1\}$, $\beta_i^{\iota+1} - \beta_i^\iota = r$ for every $i \in S_{\iota+1}$, where $\beta^\iota = \Psi(M^\iota, \pi^{b^\iota})$ and $\beta^{\iota+1} = \Psi(M^{\iota+1}, \pi^{b^{\iota+1}})$, and $r = \frac{1}{|S_{\iota+1}|} \cdot \sum_{S \subseteq \{1,...,n\}} w \cdot [\Delta_S^{\iota+1} - \Delta_S^\iota]$.*

*Proof.* Notice that for every two agents $j, k \in S_{\iota+1}$ it holds that $c(M, \pi^b, j) = c(M, \pi^b, k)$. Given Claim 4, this implies that $c(M^\iota, \pi^{b^\iota}, j) = c(M^\iota, \pi^{b^\iota}, k)$. Notice also that $S \cup \{j\} \cap C_1 \neq S_{\iota+1}$ and $S \cup \{k\} \cap C_1 \neq S_{\iota+1}$ for every $S \subseteq \{1,...,n\} \setminus \{j, k\}$. Given Observation 1, this implies that $\Delta_{S \cup \{j\}}^{\iota+1} - \Delta_{S \cup \{j\}}^\iota = \Delta_{S \cup \{k\}}^{\iota+1} - \Delta_{S \cup \{k\}}^\iota = 0$ for every $S \subseteq \{1,...,n\} \setminus \{j, k\}$. Furthermore, from Claim 4 we have that $c(M, \pi^b, i) = c(M^\iota, \pi^{b^\iota}, i) = c(M^{\iota+1}, \pi^{b^{\iota+1}}, i)$ for every $i$, and thus Lemma

---

[11]Although it is not needed for the proofs of Claims 6 and 7, we mention that $\Delta_S^{\iota+1} = \Delta_S^\iota + \Delta_{S_{\iota+1}}$ for every $S$ such that $S \cap C_1 = S_{\iota+1}$.

6 applies, and for every $j, k \in S_{\iota+1}$ we have that $\beta_j{}^{\iota+1} - \beta_j{}^{\iota} = \beta_k{}^{\iota+1} - \beta_k{}^{\iota} = r$, where $r$ is some constant. Notice that:

$$r = \frac{1}{|S_{\iota+1}|} \cdot \sum_{i \in S_{\iota+1}} \beta_i{}^{\iota+1} - \beta_i{}^{\iota}.$$

By using Claim 5 and Claim 6, we have that $r = \frac{1}{|S_{\iota+1}|} \cdot \sum_{i \in \{1,...,n\}} \beta_i{}^{\iota+1} - \beta_i{}^{\iota}$. Given that $\Psi$ is assumed to satisfy $\mathcal{R}_{AE}$ (average efficiency), this implies that:

$$r = \frac{1}{|S_{\iota+1}|} \cdot \sum_{i \in \{1,...,n\}} \beta_i{}^{\iota+1} - \beta_i{}^{\iota} = \frac{1}{|S_{\iota+1}|} \cdot \sum_{S \subseteq \{1,...,n\}} w \cdot \left[ \Delta_S^{\iota+1} - \Delta_S^{\iota} \right],$$

and hence $\beta_i{}^{\iota+1} - \beta_i{}^{\iota} = r = \frac{1}{|S_{\iota+1}|} \cdot \sum_{S \subseteq \{1,...,n\}} w \cdot \left[ \Delta_S^{\iota+1} - \Delta_S^{\iota} \right]$ for every $i \in S_{\iota+1}$. $\qquad \square$

## J  Proofs of the Results from Section 4

This section of the appendix contains the proofs of the results from Section 3, in particular: Proposition 4, Proposition 5, and Theorem 3.

### J.1  Proof of Proposition 4

**Proposition 4.** *Let $\widehat{\pi}^b$ be a solution to the optimization problem $\max_{\pi \in \mathcal{P}(\pi^b)} J(\pi)$. Then $\widehat{\Psi}_{SV,V}(M, \mathcal{P}(\pi^b)) = \Psi_{SV}(M, \widehat{\pi}^b)$ satisfies $\mathcal{R}_V$ (validity).*

*Proof.* In the setting of interest, $P(\pi^b)$ is consistent with $\pi^b$, that is $\pi^b \in P(\pi^b)$, and hence $J(\widehat{\pi}^b) \geq J(\pi^b)$. By Theorem 1, the blame attribution method $\Psi_{SV}$ satisfies property $\mathcal{R}_E$ (efficiency), which implies that $\sum_{i=1}^n \widehat{\beta}_i = J(\pi^*) - J(\widehat{\pi}^b)$, where $\widehat{\beta} = \Psi_{SV}(M, \widehat{\pi}^b)$. This implies:

$$J(\widehat{\pi}^b) \geq J(\pi^b) \Rightarrow$$
$$\Rightarrow J(\pi^*) - J(\widehat{\pi}^b) \leq J(\pi^*) - J(\pi^b) \Rightarrow$$
$$\Rightarrow J(\pi^*) - J(\widehat{\pi}^b) \leq \Delta \Rightarrow$$
$$\Rightarrow \sum_{i=1}^n \widehat{\beta}_i \leq \Delta.$$

Therefore, $\widehat{\Psi}_{SV,V}$ satisfies property $\mathcal{R}_V$ (validity). $\qquad \square$

### J.2  Proof of Proposition 5

**Proposition 5.** *Let $\beta_i^i$ be the minimum value of the objective in (P2). Then $\widehat{\Psi}_{SV,BC}(M, \mathcal{P}(\pi^b)) = (\beta_1^1, ..., \beta_n^n)$ satisfies $\mathcal{R}_V$ (validity) and $\mathcal{R}_{BC}(\Psi_{SV})$ (Blackstone consistency w.r.t. $\Psi_{SV}(M, \pi^b)$).*

*Proof.* Let $\beta = \Psi_{SV}(M, \pi^b)$. Given Eq. (1), $\beta_i^i$ being the minimum value of the objective in (P2) implies that $\beta_i^i = \min_{\pi \in \mathcal{P}(\pi^b)} \beta_i{}^\pi$ s.t. $\beta^\pi = \Psi_{SV}(M, \pi)$. In the setting of interest, $P(\pi^b)$ is consistent with $\pi^b$, that is $\pi^b \in P(\pi^b)$, which implies that $\beta_i^i = \min_{\pi \in \mathcal{P}(\pi^b)} \beta_i{}^\pi \leq \beta_i$. Therefore, $\widehat{\Psi}_{SV,BC}(M, \mathcal{P}(\pi^b))$ satisfies $\mathcal{R}_{BC}(\Psi_{SV})$ (Blackstone consistency w.r.t. $\Psi_{SV}(M, \pi^b)$). Furthermore, by applying the same reasoning to all agents, we obtain $\sum_{i \in \{1,...,n\}} \beta_i^i \leq \sum_{i \in \{1,...,n\}} \beta_i$. Given Theorem 1, this implies $\sum_{i \in \{1,...,n\}} \beta_i^i \leq \Delta$, and hence $\widehat{\Psi}_{SV,BC}(M, \mathcal{P}(\pi^b))$ also satisfies $\mathcal{R}_V$ (validity). $\qquad \square$

### J.3  Proof of Theorem 3

**Theorem 3.** *Consider $\widehat{\Psi}$ and $\Psi$ s.t. $\left\| \widehat{\Psi}(M, \mathcal{P}(\pi^b)) - \Psi(M, \pi^b) \right\|_1 \leq \epsilon$ for any $M$, $\pi^b$, and $\mathcal{P}(\pi^b)$. Then if $\Psi$ satisfies a property $\mathcal{R} \in \{\mathcal{R}_V, \mathcal{R}_E, \mathcal{R}_R, \mathcal{R}_S, \mathcal{R}_I, \mathcal{R}_{AE}\}$, $\widehat{\Psi}$ satisfies $\epsilon$-$\mathcal{R}$. Moreover, if $\Psi$ satisfies a property $\mathcal{R} \in \{\mathcal{R}_{CM}, \mathcal{R}_{PerM}, \mathcal{R}_{cPerM}, \mathcal{R}_{cParM}, \mathcal{R}_{RcParM}\}$, $\widehat{\Psi}$ satisfies $2\epsilon$-$\mathcal{R}$.*

*Proof.* We prove the implication for each property $\mathcal{R}$:

- $\underline{\mathcal{R}_V \text{ (validity)}}$: Let $\beta = \Psi(M, \pi^b)$ and $\widehat{\beta} = \widehat{\Psi}(M, \mathcal{P}(\pi^b))$. If $\Psi$ satisfies $\mathcal{R}_V$,

$$\left\| \widehat{\beta} - \beta \right\|_1 \leq \epsilon \Rightarrow \sum_{i=1}^n |\widehat{\beta}_i - \beta_i| \leq \epsilon \Rightarrow \left| \sum_{i=1}^n \widehat{\beta}_i - \beta_i \right| \leq \epsilon \Rightarrow$$

$$\Rightarrow \sum_{i=1}^n \widehat{\beta}_i \leq \sum_{i=1}^n \beta_i + \epsilon \Rightarrow \sum_{i=1}^n \widehat{\beta}_i \leq \Delta + \epsilon,$$

  and hence $\widehat{\Psi}$ satisfies $\epsilon\text{-}\mathcal{R}_V$.
- $\underline{\mathcal{R}_E \text{ (efficiency)}}$: Let $\beta = \Psi(M, \pi^b)$ and $\widehat{\beta} = \widehat{\Psi}(M, \mathcal{P}(\pi^b))$. If $\Psi$ satisfies $\mathcal{R}_E$,

$$\left\| \widehat{\beta} - \beta \right\|_1 \leq \epsilon \Rightarrow \sum_{i=1}^n |\widehat{\beta}_i - \beta_i| \leq \epsilon \Rightarrow \left| \sum_{i=1}^n \widehat{\beta}_i - \beta_i \right| \leq \epsilon \Rightarrow$$

$$\Rightarrow \left| \sum_{i=1}^n \widehat{\beta}_i - \Delta \right| \leq \epsilon,$$

  and hence $\widehat{\Psi}$ satisfies $\epsilon\text{-}\mathcal{R}_E$.
- $\underline{\mathcal{R}_R \text{ (rationality)}}$: Let $\beta = \Psi(M, \pi^b)$ and $\widehat{\beta} = \widehat{\Psi}(M, \mathcal{P}(\pi^b))$. If $\Psi$ satisfies $\mathcal{R}_R$,

$$\left\| \widehat{\beta} - \beta \right\|_1 \leq \epsilon \Rightarrow \sum_{i=1}^n |\widehat{\beta}_i - \beta_i| \leq \epsilon \Rightarrow \sum_{i \in S} |\widehat{\beta}_i - \beta_i| \leq \epsilon \Rightarrow$$

$$\Rightarrow \left| \sum_{i \in S} \widehat{\beta}_i - \beta_i \right| \leq \epsilon \Rightarrow \sum_{i \in S} \widehat{\beta}_i \leq \sum_{i \in S} \beta_i + \epsilon \Rightarrow \sum_{i \in S} \widehat{\beta}_i \leq \Delta_S + \epsilon,$$

  and hence $\widehat{\Psi}$ satisfies $\epsilon\text{-}\mathcal{R}_R$.
- $\underline{\mathcal{R}_S \text{ (symmetry)}}$: Let $\beta = \Psi(M, \pi^b)$ and $\widehat{\beta} = \widehat{\Psi}(M, \mathcal{P}(\pi^b))$. If $\Psi$ satisfies $\mathcal{R}_S$,

$$\left\| \widehat{\beta} - \beta \right\|_1 \leq \epsilon \Rightarrow \sum_{i=1}^n |\widehat{\beta}_i - \beta_i| \leq \epsilon \Rightarrow |\widehat{\beta}_i - \beta_i| + |\widehat{\beta}_j - \beta_j| \leq \epsilon \Rightarrow$$

$$\Rightarrow \widehat{\beta}_i - \beta_i - \widehat{\beta}_j + \beta_j \leq \epsilon \Rightarrow \widehat{\beta}_i - \widehat{\beta}_j \leq \epsilon \tag{r1}$$

  and

$$\left\| \widehat{\beta} - \beta \right\|_1 \leq \epsilon \Rightarrow \sum_{i=1}^n |\widehat{\beta}_i - \beta_i| \leq \epsilon \Rightarrow |\widehat{\beta}_i - \beta_i| + |\widehat{\beta}_j - \beta_j| \leq \epsilon \Rightarrow$$

$$\Rightarrow -\widehat{\beta}_i + \beta_i + \widehat{\beta}_j - \beta_j \leq \epsilon \Rightarrow -\widehat{\beta}_i + \widehat{\beta}_j \leq \epsilon. \tag{r2}$$

  From (r1) and (r2), we have $|\widehat{\beta}_i - \widehat{\beta}_j| \leq \epsilon$, and hence $\widehat{\Psi}$ satisfies $\epsilon\text{-}\mathcal{R}_S$.
- $\underline{\mathcal{R}_I \text{ (invariance)}}$: Let $\beta = \Psi(M, \pi^b)$ and $\widehat{\beta} = \widehat{\Psi}(M, \mathcal{P}(\pi^b))$. If $\Psi$ satisfies $\mathcal{R}_I$,

$$\left\| \widehat{\beta} - \beta \right\|_1 \leq \epsilon \Rightarrow \sum_{i=1}^n |\widehat{\beta}_i - \beta_i| \leq \epsilon \Rightarrow |\widehat{\beta}_i - \beta_i| \leq \epsilon \Rightarrow$$

$$\Rightarrow |\widehat{\beta}_i| \leq \epsilon \Rightarrow \widehat{\beta}_i \leq \epsilon,$$

  and hence $\widehat{\Psi}$ satisfies $\epsilon\text{-}\mathcal{R}_I$.
- $\underline{\mathcal{R}_{AE} \text{ (average efficiency)}}$: Let $\beta = \Psi(M, \pi^b)$ and $\widehat{\beta} = \widehat{\Psi}(M, \mathcal{P}(\pi^b))$. If $\Psi$ satisfies $\mathcal{R}_{AE}$,

$$||\widehat{\beta} - \beta||_1 \leq \epsilon \Rightarrow \sum_{i=1}^n |\widehat{\beta}_i - \beta_i| \leq \epsilon \Rightarrow \left| \sum_{i=1}^n \widehat{\beta}_i - \beta_i \right| \leq \epsilon \Rightarrow$$

$$\Rightarrow \left| \sum_{i=1}^n \widehat{\beta}_i - \sum_{S \subseteq \{1,\ldots,n\}} \frac{1}{2^n - 1} \cdot \Delta_S \right| \leq \epsilon,$$

  and hence $\widehat{\Psi}$ satisfies $\epsilon\text{-}\mathcal{R}_{AE}$.

- $\mathcal{R}_{CM}$ (contribution monotonicity) and $\mathcal{R}_{cParM}$ (c-participation monotonicity): Let $\beta^1 = \Psi(M^1, \pi^{b^1})$, $\beta^2 = \Psi(M^2, \pi^{b^2})$, $\widehat{\beta}^1 = \widehat{\Psi}(M^1, \mathcal{P}(\pi^{b^1}))$ and $\widehat{\beta}^2 = \widehat{\Psi}(M^2, \mathcal{P}(\pi^{b^2}))$. To show that $\Psi$ satisfying $\mathcal{R}_{CM}$ (resp. $\mathcal{R}_{cParM}$) implies that $\widehat{\Psi}$ satisfies $2\epsilon$-$\mathcal{R}_{CM}$ (resp. $2\epsilon$-$\mathcal{R}_{cParM}$), it suffices to show that $\beta_i^1 - \beta_i^2 \geq 0$ implies $\widehat{\beta}_i^1 \geq \widehat{\beta}_i^2 - 2\epsilon$. Let $\beta_i^1 - \beta_i^2 \geq 0$. We have

$$\left\| \widehat{\beta}^1 - \beta^1 \right\|_1 \leq \epsilon \Rightarrow \sum_{i=1}^{n} |\widehat{\beta}_i^1 - \beta_i^1| \leq \epsilon \Rightarrow |\widehat{\beta}_i^1 - \beta_i^1| \leq \epsilon \Rightarrow$$

$$\Rightarrow \beta_i^1 - \widehat{\beta}_i^1 \leq \epsilon \tag{r3}$$

and

$$\left\| \widehat{\beta}^2 - \beta^2 \right\|_1 \leq \epsilon \Rightarrow \sum_{i=1}^{n} |\widehat{\beta}_i^2 - \beta_i^2| \leq \epsilon \Rightarrow |\widehat{\beta}_i^2 - \beta_i^2| \leq \epsilon \Rightarrow$$

$$\Rightarrow \widehat{\beta}_i^2 - \beta_i^2 \leq \epsilon. \tag{r4}$$

By adding (r3) and (r4), we obtain

$$\beta_i^1 - \widehat{\beta}_i^1 + \widehat{\beta}_i^2 - \beta_i^2 \leq 2\epsilon \Rightarrow \widehat{\beta}_i^1 \geq \widehat{\beta}_i^2 - 2\epsilon,$$

and hence $\widehat{\Psi}$ satisfies $2\epsilon$-$\mathcal{R}_{CM}$ (resp. $2\epsilon$-$\mathcal{R}_{cParM}$).

- $\mathcal{R}_{PerM}$ (performance monotonicity) and $\mathcal{R}_{cPerM}$ (c-performance monotonicity): Let $\beta = \Psi(M, (\pi_i, \pi^b_{-i}))$, $\beta' = \Psi(M, (\pi_i', \pi^b_{-i}))$, $\widehat{\beta} = \widehat{\Psi}(M, \mathcal{P}((\pi_i, \pi^b_{-i})))$ and $\widehat{\beta}' = \widehat{\Psi}(M, \mathcal{P}((\pi_i', \pi^b_{-i})))$. To show that $\Psi$ satisfying $\mathcal{R}_{PerM}$ (resp. $\mathcal{R}_{cPerM}$) implies that $\widehat{\Psi}$ satisfies $2\epsilon$-$\mathcal{R}_{PerM}$ (resp. $2\epsilon$-$\mathcal{R}_{cPerM}$), it suffices to show that $\beta_i \geq \beta_i'$ implies $\widehat{\beta}_i \geq \widehat{\beta}_i' - 2\epsilon$. Let $\beta_i \geq \beta_i'$. We have

$$\left\| \widehat{\beta} - \beta \right\|_1 \leq \epsilon \Rightarrow \sum_{i=1}^{n} |\widehat{\beta}_i - \beta_i| \leq \epsilon \Rightarrow |\widehat{\beta}_i - \beta_i| \leq \epsilon \Rightarrow$$

$$\Rightarrow \beta_i - \widehat{\beta}_i \leq \epsilon \tag{r5}$$

and

$$\left\| \widehat{\beta}' - \beta' \right\|_1 \leq \epsilon \Rightarrow \sum_{i=1}^{n} |\widehat{\beta}_i' - \beta_i'| \leq \epsilon \Rightarrow |\widehat{\beta}_i' - \beta_i'| \leq \epsilon \Rightarrow$$

$$\Rightarrow \widehat{\beta}_i' - \beta_i' \leq \epsilon. \tag{r6}$$

By adding (r5) and (r6), we obtain

$$\beta_i - \widehat{\beta}_i + \widehat{\beta}_i' - \beta_i' \leq 2\epsilon \Rightarrow \widehat{\beta}_i \geq \widehat{\beta}_i' - 2\epsilon,$$

and hence $\widehat{\Psi}$ satisfies $2\epsilon$-$\mathcal{R}_{PerM}$ (resp. $2\epsilon$-$\mathcal{R}_{cPerM}$).

- $\mathcal{R}_{RcParM}$ (relative c-participation monotonicity): Let $\beta^1 = \Psi(M^1, \pi^{b^1})$, $\beta^2 = \Psi(M^2, \pi^{b^2})$, $\widehat{\beta}^1 = \widehat{\Psi}(M^1, \mathcal{P}(\pi^{b^1}))$ and $\widehat{\beta}^2 = \widehat{\Psi}(M^2, \mathcal{P}(\pi^{b^2}))$. To show that $\Psi$ satisfying $\mathcal{R}_{RcParM}$ implies that $\widehat{\Psi}$ satisfies $2\epsilon$-$\mathcal{R}_{RcParM}$, it suffices to show that $\beta_j^1 - \beta_j^2 \geq \beta_k^1 - \beta_k^2$ implies $\widehat{\beta}_j^1 - \widehat{\beta}_j^2 \geq \widehat{\beta}_k^1 - \widehat{\beta}_k^2 - 2\epsilon$. Let $\beta_j^1 - \beta_j^2 \geq \beta_k^1 - \beta_k^2$. We have

$$\left\| \widehat{\beta}^1 - \beta^1 \right\|_1 \leq \epsilon \Rightarrow \sum_{i=1}^{n} |\widehat{\beta}_i^1 - \beta_i^1| \leq \epsilon \Rightarrow |\widehat{\beta}_j^1 - \beta_j^1| + |\widehat{\beta}_k^1 - \beta_k^1| \leq \epsilon \Rightarrow$$

$$\Rightarrow \beta_j^1 - \widehat{\beta}_j^1 - \beta_k^1 + \widehat{\beta}_k^1 \leq \epsilon \tag{r7}$$

and

$$\left\| \widehat{\beta}^2 - \beta^2 \right\|_1 \leq \epsilon \Rightarrow \sum_{i=1}^{n} |\widehat{\beta}_i^2 - \beta_i^2| \leq \epsilon \Rightarrow |\widehat{\beta}_j^2 - \beta_j^2| + |\widehat{\beta}_k^2 - \beta_k^2| \leq \epsilon \Rightarrow$$

$$\Rightarrow -\beta_j{}^2 + \widehat{\beta_j}{}^2 + \beta_k{}^2 - \widehat{\beta_k}{}^2 \leq \epsilon. \tag{r8}$$

By adding (r7) and (r8), we obtain

$$\beta_j{}^1 - \widehat{\beta_j}{}^1 - \beta_k{}^1 + \widehat{\beta_k}{}^1 - \beta_j{}^2 + \widehat{\beta_j}{}^2 + \beta_k{}^2 - \widehat{\beta_k}{}^2 \leq 2\epsilon \Rightarrow$$
$$\Rightarrow \widehat{\beta_j}{}^1 - \widehat{\beta_j}{}^2 \geq \widehat{\beta_k}{}^1 - \widehat{\beta_k}{}^2 - 2\epsilon,$$

and hence $\widehat{\Psi}$ satisfies $2\epsilon$-$\mathcal{R}_{RcParM}$.

$\square$