# OpenReview forum: "On Blame Attribution for Accountable Multi-Agent Sequential Decision Making"
_NeurIPS.cc/2021/Conference — NeurIPS 2021 Poster_

### Official Review · Reviewer_vy59 · 2021-07-16

**Rating:** 7
**Confidence:** 2

**Summary:**

In the context of MMDPs, this work presents an overview of blame attribution methods from the game theoretic literature, together with a study of desirable properties. It then contributes a novel blame attribution method to mitigate some of the identified issues. All the methods are compared in an empirical study and analysed in terms of robustness in the face of uncertainty.



**Ethical Concerns:**

No ethical concerns.

**Limitations And Societal Impact:**

No statement for these aspects is present.

**Main Review:**

The paper is nicely written. I could follow along even without being familiar with the literature on blame attribution.

My first major comment for this work: the paper is written more as a survey, will full overview of approaches and properties, so it is difficult for readers to distinguish what the contribution and the novel concepts are, especially if they are not familiar with all the elements. There is no delimitation between existing literature and newly introduced elements. This chips away from the clarity and quality of the work.

As far as I can tell, the work is technically sound, with no evident issues.

A second major comment concerns the experimental environments. Can the authors explain the choice of the settings? It was especially difficult to grasp Environment 1, and the reasoning behind the manner with which the setting is extended to multiple agents. Is A1 meant to rectify potential mistakes of A2? It was also not clear how the policies of the agents function. For example 'The personal policy of A2 takes an optimal action with a state dependent probability; otherwise it follows an optimal policy that sets the cost of H cells to the cost of F cells'. Do you mean that the optimal policies assumes that H and F cells have the same cost?

All in all, it was an interesting study, my only thought is that this work could have a better impact in a longer format, that would also incorporate all the numerous details from the appendix.


Minor comments:

Lines 115 vs. 119: inconsistent $\mathbb{R}$
Line 103: the expected return is what is presented there

---Post-rebuttal---
Thank you for addressing the raised concerns and for all the clarifications. I will raise my score accordingly.

**Time Spent Reviewing:**

4

---

> ### Author Response · Authors · 2021-08-09
> **Response to Reviewer vy59**
>
> Thank you for your comments and valuable suggestions. We are glad to see that you find this paper interesting and nicely written. Please find below our response to your comments.
>
> -----
>
> **Contributions and novel concepts**
>
> While this paper utilizes existing concepts from prior work (e.g., from the cost sharing literature) in the context of multi-agent sequential decision making, it also contributes novel concepts. We highlight the following contributions and the novel concepts introduced in the paper:
> - Formalizing the set of desirable properties of a blame attribution method in the context of multi-agent sequential decision making. These include properties from the cost sharing literature, but also novel properties: *performance monotonicity* and *Blackstone consistency*, which we believe are important for accountable decision making.
> - Characterizing blame attribution methods under this set of properties. In addition to well-known blame attribution methods, such as Shapley value (SV), we also study a novel blame attribution method, which we call *average participation* (AP). This blame attribution method is motivated by our characterization results which show that there are trade-offs between different desirable properties.
> - Studying and characterizing the blame attribution methods under uncertainty, i.e., when they do not have direct access to the agents' behavior policies.
>
> -----
>
> **Choice of experimental environments**
>
> These are very useful remarks. Reviewer iKJr had a similar question regarding the experimental environments, so please also read our response to Reviewer iKJr (*Experimental evaluation*). We briefly reiterate the main points below. We selected two environments from prior work (Voloshin et al. *Empirical study of off-policy policy evaluation for reinforcement learning*, reference [52]), and modified them to have a multi-agent structure. We specifically selected these two (modified) environments for the following reasons:
>
> - In the *Gridworld* environment, agent $A_1$ selects its policy based on the model of agent $A_2$ (i.e., $A_1$ is trained to best respond to its model of $A_2$). Since we can control the correctness of $A_1$'s model of $A_2$, we can also test whether best responding to $A_2$ is the optimal choice for $A_1$. In particular, if $A_1$ minimizes its blame by best responding to the correct model of $A_2$, then the blame attribution method is performance incentivizing. This in turn means that the *Gridworld* environment is suitable for testing the *performance monotonicity* property.
>
> - In the *Graph* environment, agents need to achieve a higher degree of coordination in order to change the outcome that occurs under the baseline (behavior) policies. In particular, for most of the cases, an agent can't improve the joint performance by unilaterally changing its policy and instead needs to coordinate with other agents. Therefore, the *Graph* environment is suitable for evaluating blame attribution methods when the outcome can't be influenced by a single agent, i.e., for testing whether blame attribution methods incorporate more nuanced counterfactual reasoning.
>
> -----
>
> **Multi-agent version of Environment 1 (Gridworld)**
>
> We are happy to provide more details regarding Environment 1 (Gridworld). The single-agent version of the *Gridworld* environment consists of a (possibly) sub-optimal agent $A_2$ who controls the movement of the actor. In our multi-agent version of this environment, we have an additional agent, agent $A_1$, who can intervene at a certain cost (per intervention) and override $A_2$'s policy. The two agents select actions simultaneously. If $A_1$ intervenes, the actor takes the action that an optimal agent would select (in the single-agent mode). If $A_1$ does not intervene, the actor takes the action that $A_2$ selects. As we mentioned above, $A_1$ is trained to best respond to its model of $A_2$; ideally, $A_1$ would rectify potential mistakes of $A_2$ that could inflict cost greater than that  of intervention, but its model of $A_2$ may not be accurate, so $A_1$ might not optimally intervene. At each state, $A_2$ follows a policy that is a mixture of an optimal single-agent policy for correctly specified costs and a single-agent policy that is optimal but for a misspecified cost of the $H$ cells (in this case, the cost of the $H$ cells is set to the cost of the $F$ cells).
>
> -----
>
> **Limitations and societal impact**
>
> We didn't add a separate section/paragraph that discusses limitation and societal impact, but Section 6 includes the discussion about these aspects.
>
> -----
>
> Thank you again for your helpful comments. Please let us know if you have any additional questions or comments.

---

### Official Review · Reviewer_4AjN · 2021-07-16

**Rating:** 7
**Confidence:** 4

**Summary:**

This paper furthers the study of blame in accountability in cooperative multi-agent systems.  They provide many natural desiderata, some natural attribution techniques, and characterizations of how these relate, as well as how they relate to standard game theory ideas.

**Limitations And Societal Impact:**

A limitation that you mentioned was the negative externality of unjustly blaming someone, but it is also worth mentioning the negative externality of not blaming someone when you should and thus incentivizing bad behaviors.

**Main Review:**

I enjoyed this paper, and thought it was generally well written and complete.  My main substantive concern was that many of the desiderata are directly consequentialist in the sense that you can only be to blame if the event would not have happened if you had changed your action.  It seems like this deviates from intuitions around blame; for instance if a pilot took a nap on the job, but the system malfunctioned in a way that would have been unrecoverable even if they were awake, many would still want to blame the pilot for not satisfying the normative commitments we expect of a pilot.  This notion of blame would be more deontological, and would certainly violate many of these desiderata, but is closer to what people do.  The phrase comes to mind "no one was fired for buying ibm"; the claim isn't that ibm gives the highest utility, it's that it is a choice that is hard to blame someone for.

Additionally, I think there is a more causal objection to the desiderata.  You may want to blame the pilot because it would have been their fault if something else where different.  That is, even though nothing they could do would have changed anything, if the autopilot system had worked, or had failed differently, the pilot would have been able to change the outcome.  In a sense this is blame two counterfactual changes away from reality rather than one.

Flagging these kinds of limitations early could further strengthen this work, as it would clearly set out the domain of inquiry.

On an organizational note, I think that the paper would be greatly improved if the experimental environments were introduced early and used as running examples.  By the time we got to the discussion of the behavior fo the rules in the environments I had forgotten most of the rules, but this would not have been the case if the behavior in the environments where discussed more gradually. It would have also made the rules more intuitive.

minor comment:
* 152 should be "does not assign less blame for a policy with at least as much performance"

------
**Update to Review**

I appreciate the interesting comments and clarifications both in the response to this review and to the other reviews.  I believe incorporating the proposed changes will be valuable to readers and will improve the paper.


**Time Spent Reviewing:**

4

---

> ### Author Response · Authors · 2021-08-09
> **Response to Reviewer 4AjN**
>
> Thank you for your comments and valuable suggestions. We are happy to see that you find this paper enjoyable to read and complete. Please find below our response to your comments.
>
> -----
>
> **Deontological arguments**
>
>
> These are excellent remarks. The problem of assigning blame is inherently multi-dimensional and can be viewed through both deontological and consequentialist lenses. For example, research in, e.g., program verification often focuses on a deontological approach on verifying whether a program adheres to a given set of rules. In contrast, we study blame attribution approaches that are more related to consequentialist reasoning. Our intention is not to replace deontological approaches, but to complement them.
> However, we agree that the list of desirable properties described in our work can be extended to include principles that are not solely focused on immediate consequences of actions, but also take a deontological perspective.
>
>
> -----
>
> **Incorporating more nuanced counterfactual reasoning**
>
> Regarding your comment about the need of incorporating more nuanced counterfactual reasoning, e.g.,
>
> > In a sense this is blame two counterfactual changes away from reality rather than one.
>
> We agree that this is an important consideration. Our framework can actually incorporate this consideration if we adequately model all the agencies in the system. In your example, if we model the autopilot system as a separate agent, the pilot would be assigned a positive degree of blame. More precisely and focusing on the studied blame attribution methods, marginal contribution (MC) and max-efficient rationality (MER) methods would still assign zero blame to the pilot (since the pilot can't avoid the accident by themselves), but the other three methods would assign positive blame (assuming that a better outcome is reached if the autopilot system acts differently and the pilot is awake). An interesting direction for future work would be to develop tools that would help us in identifying relevant agencies in the system of interest and entities responsible for them.
>
>
> -----
>
> **Negative externality of not blaming someone**
>
> This is an interesting point, which also explains why trivial blame attribution methods (e.g., assigning 0 blame to everyone) are not interesting. We will add this remark to the paper.
>
>
> -----
>
> Thank you for your other comments as well, including the organizational note. Your feedback will help us improve the paper. Please let us know if you have any additional questions or comments.

---

### Official Review · Reviewer_iKJr · 2021-07-16

**Rating:** 7
**Confidence:** 4

**Summary:**

This paper provides a formal definition of blame for MA MDPs, a set of properties (many from the cooperative game theory literature, but also some for this setting) and an analysis of several methods (both existing methods like Shapley value and a new method that has properties Shapley does not have that are useful in this context. They also discuss uncertainty-oriented extensions to these ideas, and provide an experimental evaluation comparing several of the existing and proposed blame attribution methods.

**Limitations And Societal Impact:**

yes; this is somewhat a core idea in the paper.

**Main Review:**

Originality: Are the tasks or methods new? Is the work a novel combination of well-known techniques? (This can be valuable!) Is it clear how this work differs from previous contributions? Is related work adequately cited?

The work is of course based in traditional cooperative game theory, but I believe it is new in this setting, with extensions to those concepts fitting for the setting. Previous work is well-cited.

Quality: Is the submission technically sound? Are claims well supported (e.g., by theoretical analysis or experimental results)? Are the methods used appropriate? Is this a complete piece of work or work in progress? Are the authors careful and honest about evaluating both the strengths and weaknesses of their work?

In general I think things are well supported (see clarity issues below), and just about what one would expect for this kind of paper. I’d like to see a little on the complexity of the new method (since Shapley is unusable in practice for this reason). Examples would be even better.

The experimental section is dense and a little unmotivated (or unclearly motivated). It is not always clear what the experiments are (the domains are reasonably described but I’m still a bit unclear on the policies of the participants and how they were manipulate, and what is being shown on the graphs. At a high level, you would be better to describe *how* you intend to evaluate, how the reader should interpret the results, and then finally show the evaluations, since what is being evaluated here is (rightly!) quite different from the typical NeurIPS paper showing the performance of learning algorithm X.

Clarity: Is the submission clearly written? Is it well organized? (If not, please make constructive suggestions for improving its clarity.) Does it adequately inform the reader? (Note that a superbly written paper provides enough information for an expert reader to reproduce its results.)

In general the paper is quite clear (at least, to someone familiar with the traditional cooperative game theory ideas). However, probably due to space issues, I found the run up to the new method in 3.4 to be extremely dense with very little glossing explanation or examples. The high level idea makes sense but not sure I could re-implement. Same for section 4 (where I am less familiar with current approaches to reasoning over uncertain policies). The notational density also starts to wear the reader down (can’t keep the acronyms all straight…need a little cheat sheet to cut out and place by the paper!). Also, comments on experiments above.

Significance: Are the results important? Are others (researchers or practitioners) likely to use the ideas or build on them? Does the submission address a difficult task in a better way than previous work? Does it advance the state of the art in a demonstrable way? Does it provide unique data, unique conclusions about existing data, or a unique theoretical or experimental approach?

Yes, I think this is quite a useful porting of these ideas to this setting--I imagine this could be used as a basis of several different future works.

== Thanks for the clarifications; hopefully those can be added to the final version ==

**Time Spent Reviewing:**

2

---

> ### Author Response · Authors · 2021-08-09
> **Response to Reviewer iKJr**
>
> Thank you for your comments and valuable suggestions. We are glad to see that you find this paper interesting and its contributions significant. Please find below our response to your comments.
>
> -----
>
> **Complexity of the new method**
>
>
> The average participation (AP) method (a novel blame attribution method) is similar to Shapley value in terms of computational complexity. As it is the case with Shapley value, evaluating the exact blame assignment using average participation involves computing the marginal inefficiency of each subset of agents. In general, for a large number of agents, computing the exact blame assignments is challenging as the number of subsets is exponential in the number of agents. However, note that we could potentially utilize sampling based approaches to approximate the output of the novel blame assignment method, akin to those for approximating Shapley value (e.g., see Jia et al., *Toward efficient data valuation based on Shapley value*), but adapted to the setting of interest. To account for the fact that the exact value is not obtained in this process, we could adopt the principles from Section 4 (*Blame Attribution under Uncertainty*) and, for example, consider the corresponding type of the Blackstone consistency property. In our experiments, we use a relatively small number of agents, and we think that many interesting multi-agent settings belong to this regime (e.g., human-AI collaboration). Nevertheless, there are settings in which computational complexity considerations are important (e.g., when a larger number of robots need to coordinate, as it is the case for warehouse robots), and we see this as an interesting future research direction to explore.
>
> -----
>
> **Experimental evaluation**
>
> These are very valuable suggestions. We will preface the experimental results with a paragraph explaining how we utilize the experiments to empirically evaluate the properties of different methods for blame attribution. We briefly outline the main reasons for choosing the environments from Section 5 and the motivation behind the performed experiments. First note that these two environments are based on prior work (Voloshin et al. *Empirical study of off-policy policy evaluation for reinforcement learning*, reference [52]), but we made them multi-agent. We specifically considered these two environments for the following reasons:
>
> - The two-agent *Gridworld* environment is suitable for testing the *performance monotonicity* property, which we deem important for accountability. In this environment, one agent, agent $A_1$, optimizes its policy w.r.t. the model of another agent, agent $A_2$. By controlling the correctness of its model of $A_2$, we can validate whether $A_1$ is indeed incentivized to *best respond* to the behavior of $A_2$. Namely, if $A_1$ minimizes its blame when its policy is trained to optimally respond to the correct model of $A_1$, the corresponding blame attribution method is performance incentivizing (i.e., it satisfies performance monotonicity). The results that test for performance monotonicity are presented in Figure 3a, and they show, for example, that Shapley value is not performance incentivizing.
>
> - The *Graph* environment is convenient for testing blame attribution methods when a high(er) degree of coordination among agents is needed to yield improvements over the baseline behavior. The goal here is to test more nuanced cases of blame attribution in multi-agent settings when the outcome can't be influenced by a single agent. Prior work has shown that in these cases a blame attribution method may need to account for the fact that agents need to coordinate to change the outcome (e.g., see Chockler and Halpern *Responsibility and blame: A structural-model approach* (reference [15])). The results that test for such nuances are presented in Figure 3e, and for example, they show that some methods have very low efficiency (in terms of how much blame they distribute) under these nuanced properties (e.g., max-efficient rationality (MER) and marginal contribution (MC) assign $0$ blame to agents whenever some degree of coordination is required).
>
> We also use these environments for testing how the blame attribution methods perform under uncertainty. These results are presented in Figures 3.b,c,d,f,g,h. The goal of the experiments related to uncertainty is: a) to experimentally investigate the trade-offs between some blame attribution properties  (in particular efficiency vs. validity and Blackstone consistency), b) to experimentally test which blame attribution methods are more robust to uncertainty.
>
> -----
>
> **Exposition of the results**
>
> Thank you for your remarks about the clarity of the paper. As you pointed out, due to the space limit, it is challenging to provide detailed explanations with examples for all of our results. This is particularly the case for Section 3.4 and Section 4, which are arguably the most technical sections of the paper. To alleviate your concerns and to provide more intuition behind our results, we will try to improve the exposition of the results from these two sections, and include extra discussions/explanations (e.g., in the appendix).
>
> -----
>
> Thank you again for your helpful feedback. Please let us know if you have any additional questions or comments.

---

### Author Response · Authors · 2021-08-09
**General response to reviews**

We thank the reviewers for their valuable comments and suggestions. We appreciate the positive feedback that we obtained from the reviewers. In our comments below, we respond to the reviewers' questions and comments.

---

### Decision · Program_Chairs · 2021-09-27

**Decision:**

Accept (Poster)

**Comment:**

All of the reviewers appreciated the general problem being tackled by this paper, and view it as an interesting one for multiagent systems and accountability/interpretability in AI. The axiomatic approach and the novelty and analysis of the new AP mechanism for blame attribution are both novel and interesting. The reviews express some concerns around presentation and explanation of some of the details (e.g., experiment motivation, the remarks around consequentialist vs. deontic perspective on the problem, etc.). The author response helped clarify a few of these issues, and nudged the consensus to acceptance. It is critical however that the authors make the revisions needed to address/clarify the issues raised. While this type of paper is somewhat a bit outside of the (current) set of mainstream topics for NeurIPS, the topic is of relevance to the community (and connects to interpretability and fairness, etc. that are becoming more mainstream in NeurIPS).